# Proactive Reasoning-with-Retrieval Framework for Medical Multimodal Large Language Models

## Abstract

Incentivizing the reasoning ability of Multimodal Large Language Models (MLLMs) is essential for medical applications to transparently analyze medical scans and provide reliable diagnosis. However, existing medical MLLMs rely solely on internal knowledge during reasoning, leading to hallucinated reasoning and factual inaccuracies when encountering cases beyond their training scope. Although recent Agentic Retrieval-Augmented Generation (RAG) methods elicit the medical model's proactive retrieval ability during reasoning, they are confined to unimodal LLMs, neglecting the crucial visual information during reasoning and retrieval. To this end, we propose the first **Multimodal Medical Reasoning-with-Retrieval framework, Med-RwR**, which proactively retrieves external knowledge by querying observed symptoms or domain-specific medical concepts during reasoning. Specifically, we design a two-stage reinforcement learning strategy with tailored rewards that stimulate the model to leverage both visual diagnostic findings and textual clinical information for effective retrieval. Building on this foundation, we further propose a **Confidence-Driven Image Re-retrieval (CDIR)** method for test-time scaling when low prediction confidence is detected. Evaluation on various public medical benchmarks demonstrates Med-RwR's significant improvements over baseline models, proving the effectiveness of enhancing reasoning capabilities with external knowledge integration. Furthermore, Med-RwR demonstrates remarkable generalizability to unfamiliar domains, evidenced by 8.8% performance gain on our proposed EchoCardiography Benchmark (ECBench), despite the scarcity of echocardiography data in the training corpus. Our data, model, and codes will be made publicly available.

## 1 Introduction

Reasoning has emerged as an essential capability in multimodal large language models (MLLMs), allowing the models to reveal the cognitive process of analyzing information, solving problems, and drawing conclusions. Existing works has employed techniques such as Long Chain-of-Thought Distillation (Yao et al., 2024; Dong et al., 2025; Du et al., 2025) and Reinforcement Learning (Liu et al., 2025b; Meng et al., 2025; Huang et al., 2025a) to develop multimodal reasoning models, with recent efforts extending these approaches to healthcare (Pan et al., 2025; Su et al., 2025; Mu et al., 2025).

However, ensuring the factual reliability of the reasoning process remains a significant challenge. Unreliable reasoning processes are inclined to propagate erroneous intermediate conclusions, resulting in incorrect decisions despite following solid logical structures. As shown in Figure 1 (a), a recent medical reasoning model, Chiron-o1 (Sun et al., 2025a), demonstrates logical reasoning flow but excludes the correct answer by referring to invalid diagnostic criteria (friability, infection) that are not distinguishing features. This occurs because existing medical reasoning models solely rely on *internal knowledge* during inference, making them prone to generate non-factual contents when encountering cases beyond their training scope. This in turn leads to misdiagnoses and reduces clinical trustworthiness, as exemplified in Figure 1 (a), where the model mistakenly identifies the condition as "fibroelastoma" instead of the ground-truth answer due to reliance on false evidence.

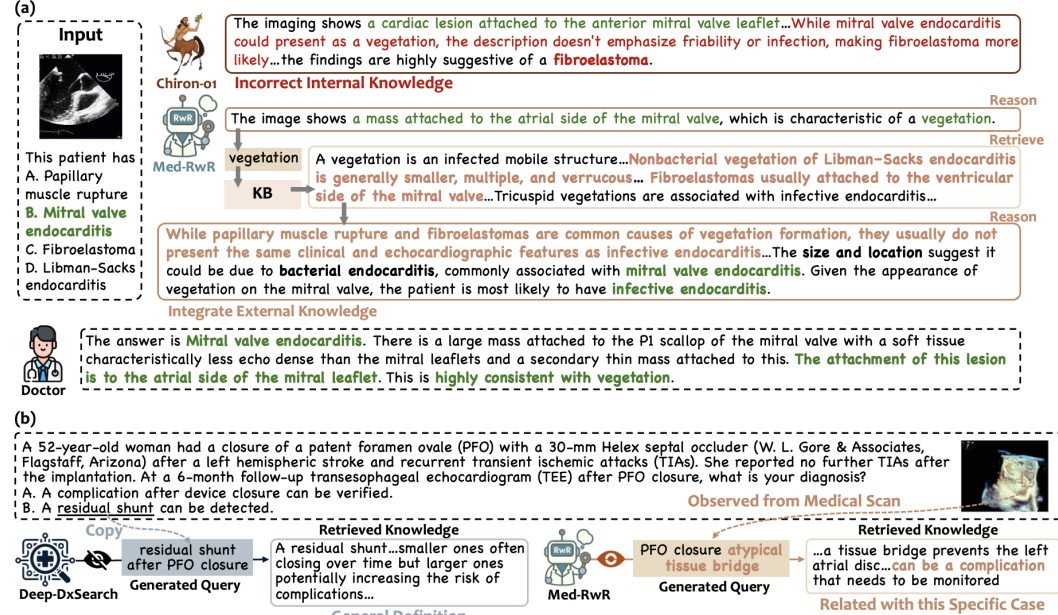

Figure 1: (a) Comparison between the generated outputs of a recent reasoning model, Chiron-o1 (Sun et al., 2025a), and the rollout process of our proposed Reasoning-with-Retrieval framework, which includes a trajectory of think, query and retrieve steps. Texts in green denote the correct statements, red indicates erroneous outputs, and brown annotates the authoritative knowledge. (b) Comparison between the effectiveness of retrieval between text-only agentic RAG system for diagnosis, Deep-DxSearch (Zheng et al., 2025), and our proposed multimodal MED-RWR.

Consequently, medical AI models should ground their reasoning in clinically authoritative evidence, rather than static and memorized knowledge alone.

While Retrieval-Augmented Generation (RAG) could expose the models to clinical evidence from external medical databases, current medical RAG models (Jeong et al., 2024; Ong et al., 2024; Zhao et al., 2025a) are constrained by static retrieval. To mitigate this issue, Agentic Search (Jin et al., 2025; Wu et al., 2025a) has emerged as a promising approach, and Ding et al. (2025); Zheng et al. (2025); Yu et al. (2025a) have adapted this framework to the medical domain, encouraging models to actively integrate *external evidence* from the knowledge base during reasoning. Nevertheless, they focus primarily on Large Language Models fed with only textual input. This is insufficient in medical diagnosis since the rich visual context may be overlooked during the proactive query generation and knowledge retrieval processes, thus fetching general definitions and failing to correspond to specific evidence in the scans. For instance, the text-only agentic RAG system, Deep-DxSearch (Zheng et al., 2025), can only execute the retrieve action based on textual hints without images as inputs. Figure 1 (b) demonstrates that Deep-DxSearch directly copies the query from the original question, resulting in retrieval of a general medical definition that lacks the discriminative features for differential diagnosis. This challenge arises because retrieving relevant knowledge requires leveraging key observations from the image, which remains largely underexplored.

To address this limitation, we propose **MED-RWR**, a multimodal **Med**ical **R**easoning-**w**ith-**R**etrieval pipeline, which incentivizes the medical MLLM to actively retrieve factual evidence during reasoning. Our core idea is to mirror how clinicians reason to diagnose: observing the medical scans while cross-referencing textual protocols and guidelines, then making final conclusions based on evidence. Specifically, we first implement a Cluster-Reconstruct-Stratify data generation process to curate a controllable environment, comprising a multimodal training dataset and a knowledge base. Subsequently, we develop a two-stage reinforcement learning (RL) framework guided by a tailored reward design aware of both visual findings and textual information, thereby ensuring the retrieved knowledge is aligned with both modalities. To augment inadequate knowledge retrieval, the trained framework further supports **C**onfidence-**D**riven **I**mage **R**e-retrieval (**CDIR**) for test-time scaling, where it can refer to similar multimodal cases when making less confident decisions. As

illustrated in Figure 1, our model first excludes the differential diseases due to incompatible size and location by querying the underlying cause of the observed phenomenon, and then confirms the correct answer via retrieved diagnosis criteria.

To demonstrate the effectiveness of our proposed method, we conduct extensive experiments on public benchmarks, and achieve an improvement of 5.1% on MedXpertQA-MM, 9.7% on MMMU-H&M, and 12.3% on MMMU-Pro-H&M, respectively, compared with the base model. Furthermore, one of the representative advantages of MED-RWR is its **generalizability to unfamiliar domains** achieved by leveraging the strong reasoning-with-retrieval capacity. To demonstrate this, we evaluate the model on echocardiography, a highly specialized medical domain that poses challenges for adaptation due to both data scarcity (less than 2% of the training corpus as shown in Figure 4) and high clinical expertise requirements. We construct EchoCardiography Benchmark (ECBench) from authoritative practice collections of multiple difficulty levels, benchmarking the model on the underrepresented multimodal echocardiography data. Our method achieves performance gains of at least 8% over other state-of-the-art methods on ECBench, demonstrating its flexibility to incorporate domain-specific external knowledge for adaptation.

To summarize, we establish **the first comprehensive multimodal Medical Reasoning-with-Retrieval framework, MED-RWR**, encompassing contributions as follows:

• We develop an environment with a curated multimodal dataset and a specialized knowledge base to train the medical MLLM to *actively ground the reasoning process in external reliable sources*, rather than depending exclusively on internal knowledge.

• We develop a two-stage reinforcement learning (RL) strategy with tailored rewards specifically designed to stimulate the medical MLLM to *be aware of both visual findings and textual contexts in retrieval* for more relevant and accurate information.

• We demonstrate the effectiveness of our approach in *enhancing medical reasoning and understanding* through extensive evaluation, achieving significant improvements on public medical multimodal benchmarks. To evaluate *generalizability to scarce medical domains*, we collect ECBench, a multimodal echocardiography benchmark. Our method demonstrates remarkable adaptation to this specialty through leveraging external expertise despite limited domain-specific training data.

## 2 RELATED WORKS

### 2.1 MEDICAL MULTIMODAL LARGE LANGUAGE MODELS

Current advances in MLLMs have accelerated their application in medical settings to assist diagnostic procedures (Wu et al., 2023; Bannur et al., 2024; Zhang et al., 2024; Wang et al., 2024; Yang et al., 2025b). However, these models directly provide diagnostic outputs without detailed explanations, making it difficult for clinicians to interpret and follow. This lack of transparency has driven the development of medical reasoning models, which explicitly output the thinking process for decision-making. Several works focus on curating reasoning chains to teach medical models to think step-by-step (Chen et al., 2024b; Huang et al., 2025b; Wu et al., 2025b), and recent efforts have applied reinforcement learning to iteratively improve reasoning quality through reward-based training (Xu et al., 2025a; Su et al., 2025; Liu et al., 2025a; Mu et al., 2025). Despite these improvements, both approaches rely heavily on the models' internal knowledge in the inference stage, which can generate reasoning processes that contain unreliable medical information especially when encountering data beyond training scope, potentially compromising clinical decision-making.

### 2.2 RETRIEVAL-AUGMENTED GENERATION FOR MEDICINE

To expose models to additional clinical evidence from external databases, Retrieval-Augmented Generation (RAG) has been applied to enhance medical models. RULE (Xia et al., 2024) addresses factual errors in medical MLLMs through calibrated retrieval context selection and preference-based fine-tuning, balancing model's reliance on inherent knowledge with external information. MMed-RAG (Xia et al., 2025b) advances this approach by introducing a more versatile multimodal RAG system that employs domain-aware retrieval and adaptive context selection. Sun et al. (2025b) augments report generation with high-quality references with a fact-aware multimodal RAG pipeline.

However, these medical RAG methods only support static retrieval, performing retrieval operations without the ability to adaptively retrieve additional information based on the model's generation process, which limits their effectiveness in handling complex medical problems.

## 2.3 AGENTIC SEARCH

To incentivize a more flexible paradigm, researchers have explored integrating retrieval into the reasoning process to enable mutual enhancement, where the retrieved information can support reasoning while reasoning helps generate more effective queries. Recent works have introduced Reinforcement Learning (RL) to stimulate this process. Search-R1 (Jin et al., 2025) and R1-Searcher (Song et al., 2025) proposed search-augmented RL training strategies to enhance retrieval-driven reasoning and decision-making. In the multimodal domain, Liu et al. (2025c); Wu et al. (2025a); Narayan et al. (2025) equip MLLMs with agentic capabilities of adaptive retrieval from Internet sources. WebWatcher (Geng et al.) enhances MLLMs' visual-language reasoning capabilities by incorporating various tools such as web search and optical character recognition (OCR).

Despite these advances, there remains a critical gap in exploring multimodal reasoning with adaptive retrieval based on both imaging and text in the medical domain, where the retrieved knowledge often supports differential diagnosis reasoning among multiple hypotheses, rather than seeking a single correct answer. This requires the model to be aware of anatomical structures and pathological patterns in the medical scans, a capability that general text-based agentic search models lack. Although Zhang et al. (2025) combines a retrieval agent with a pathology MLLM to enable agentic multimodal retrieval from pathology textbooks, its decoupled architecture and restriction to pathology-specific knowledge limit its ability to perform integrated retrieval during reasoning across diverse medical modalities. In contrast, we develop a two-stage reinforcement learning (RL) strategy with medically-tailored rewards to train the model in an end-to-end manner with diverse modalities. Our designed rewards can enable the model to leverage both visual findings and textual contexts when formulating queries, ensuring more relevant and accurate retrieved information.

## 3 METHODOLOGY

In this section, we elaborate on the proactive multimodal reasoning-with-retrieval framework. We first curate a controllable environment with a complex multimodal training dataset and a specialized knowledge base (§3.1). With the training environment prepared, we design a two-stage reinforcement learning (RL) strategy with tailored rewards to incentivize external knowledge retrieval during reasoning (§3.2). To augment inadequate information retrieval during inference, we propose a confidence-driven image re-retrieval scheme for test-time computational scaling (§3.3).

### 3.1 REASONING-WITH-RETRIEVAL ENVIRONMENT

Our training environment for facilitating reasoning-with-retrieval ability comprises a multimodal training dataset and a knowledge base carefully aligned with its scope. To construct a multimodal dataset with distinct difficulty levels for reinforcement learning, we employ a Cluster-Reconstruct-Stratify pipeline using PubMedVision (Chen et al., 2024c) as the data source. **Cluster**: Firstly, we perform K-means clustering to group all image features into 2,000 clusters to balance between diversity and efficiency , and sample representative instances near cluster centers to ensure coverage of diverse visual patterns. **Reconstruct**: We input selected images with original captions into GPT-4o to extract background knowledge, observation, analysis and conclusion. Then, we further prompt it to reformulate multi-choice questions asking for conclusions based on the background without revealing any visual observation or analysis. Illustrative examples are presented in Figure 5. **Stratify**: To assess question difficulty, we employ two MLLMs, QwenVL2.5-7B (Bai et al., 2025) and InternVL3-8B (Zhu et al., 2025) to generate 10 responses for each question and measure accuracy. We remove trivial questions with correct answers in all iterations, and stratify the remaining questions into easy and difficult categories based on accuracy thresholds. Finally, we obtain approximately 6,500 multimodal question-answer pairs of different difficulty levels for progressive curriculum training. Specifically, we manually arrange the training dataset from easier to harder based on the difficulty label. During training, the data loader processes the dataset in this sequential order, enabling the model to learn from simpler examples and then progress to more complex

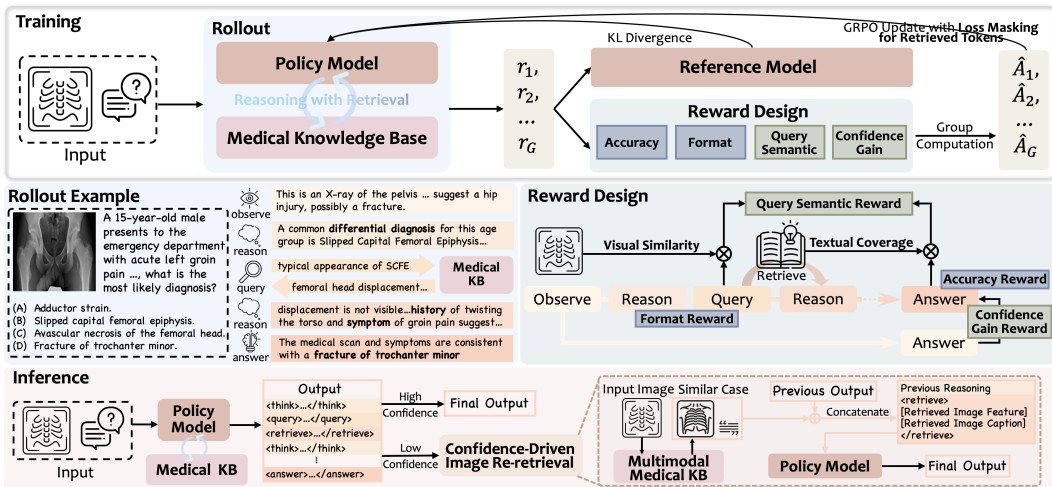

Figure 2: Overall Illustration of the Reasoning-with-Retrieval framework. During training, the model learns to generate reasoning steps while actively retrieving from a medical knowledge base. During inference, the Confidence-Driven Image Re-retrieval scheme would be triggered to augment less informative retrieval when low confidence is detected.

ones. Dataset statistics are in Appendix A.1. To support the retrieval action of our framework, we construct a medical knowledge base by extracting medical knowledge from the analytical contents generated during the training data construction process, as detailed in Appendix C.1.

### 3.2 MULTIMODAL MEDICAL REASONING-WITH-RETRIEVAL FRAMEWORK

#### 3.2.1 PROBLEM FORMULATION

We formulate medical visual question answering as a reasoning-with-retrieval framework. Specifically, the MLLM is modeled as the policy model $\pi_\theta(\cdot \mid x)$, which performs autoregressive prediction given input $x = \{i, t\}$ consisting of medical images $i$ and text prompts $t$. The model $\pi_\theta$ first outputs reasoning steps $h$ based on the multimodal input $x$ before generating the final response $y$. During reasoning, the model can decide to generate a query $q$ to retrieve relevant information $k \in \mathcal{K}$ from an external medical knowledge database $\mathcal{K}$. The retrieved knowledge $k$ and the reasoning history $h$ are then incorporated into the model's context, enabling the model to continue generation with enriched information, $\pi_\theta(\cdot \mid x, h, k)$, until reaching a final answer.

#### 3.2.2 REWARD DESIGN

To equip our proposed framework with proactive reasoning-with-retrieval ability, we design four rewards to ensure that the model learns effective retrieval strategies to seek highly relevant knowledge that aligns with the multimodal inputs.

**Retrieval Format Reward.** We first define the format reward to ensure structured output generation. The model is expected to enclose the thinking process within <think> and </think> and retrieval query within <query> and </query>. The retrieval system will search in the database $\mathcal{K}$ based on the query and return the matched information within <retrieve> and </retrieve>. The final answer is enclosed between <answer> and </answer>. The format reward $\sigma_f$ equals 1 if the output includes a thinking process, with an additional 1 when retrieval is activated and leads to a correct answer, and 0 otherwise.

**Accuracy Reward.** The accuracy reward directly evaluates whether the model's output is correct. The answer choice is extracted from the generation and compared against the ground truth. The accuracy reward $\sigma_a$ is set to 1 for correct choices and 0 for incorrect ones.

**Query Semantic Reward.** We propose the Query Semantic Reward to drive the model to generate queries with enhanced semantic alignment between both the text prompt and visual content. This

could improve the contextual relevance of the retrieval process, contributing to enhanced response accuracy. The reward consists of two components to separately address textual and visual alignment.

The first part of the reward promotes the medical-specific textual semantic alignment among the generated query, retrieved contents, and ground-truth content. Specifically, we identify medical entities from them using a named-entity recognition (NER) model (Kraljevic et al., 2021) pretrained to detect clinical concepts within the Unified Medical Language System (UMLS). The reward is then calculated as the proportion of overlap entities, thereby incentivizing the model to generate queries that drive the reasoning process toward the ground truth. We give the formal formulation of this reward as Eqn. 1 below, where $S_Q, S_K, S_G$ denote the medical entity set of generated query, retrieved information, and ground truth content.

$$\sigma_{q_{\text{text}}} = \frac{|S_Q \cap S_G|}{|S_Q|} + \frac{|S_K \cap S_G|}{|S_K|}, \tag{1}$$

The second part of the reward enhances the visual semantic alignment between the generated query and the input image. This design prevents the model from learning shortcuts (Xia et al., 2025a), i.e., querying solely based on the textual prompt while ignoring the visual information, and improves the visual relevance of the retrieved knowledge. Specifically, it maximizes the semantic similarity between the generated query and the input image within the shared embedding space, with formal formulation given as Eqn. 2, where $i, t$ denote the visual and textual input, and $f_{image}, f_{text}$ are the pretrained image and text encoders from BiomedCLIP (Zhang et al., 2023a).

$$\sigma_{q_{\text{image}}} = \frac{f_{\text{image}}(i) \cdot f_{\text{text}}(t)}{||f_{\text{image}}(i)|| \, ||f_{\text{text}}(t)||}, \tag{2}$$

Through additional visual alignment, the model is encouraged to leverage visual features during query formulation. The final query semantic reward is a linear combination of both Eqn. 1 and Eqn. 2, denoted as $\sigma_q = \sigma_{q_{\text{text}}} + \sigma_{q_{\text{image}}}$.

**Confidence Gain Reward.** To ensure that the retrieval process substantively contributes to reaching the accurate response, we establish a connection between the retrieval mechanism and the model's confidence in providing the correct answer. Intuitively, a model supported by the retrieved knowledge should exhibit higher confidence in predicting the correct answer compared to when retrieval is absent. Building on this, we propose the confidence gain reward, which quantifies the improvement in the model's confidence between its output with retrieved content $y$ and its output without retrieved content $y'$. Specifically, let the token position of <answer> be denoted as $\tau$. At the position $\tau + 1$, the model generates the answer to a multiple-choice question. To compute the confidence gain, we measure the difference in the predicted log probabilities of the ground-truth answer token before and after the retrieval process. Eqn. 3 gives the formulation of confidence gain reward, where $h$ denotes the previous reasoning history, $k$ denotes the retrieved information.

$$\sigma_c = \log \frac{\pi_\theta((y')_{\tau'+1}^{\text{answer}} \mid x, y'_{\leq \tau'}, h, k)}{\pi_\theta(y_{\tau+1}^{\text{answer}} \mid x, y_{\leq \tau})}, \tag{3}$$

**Overall Rewards.** Finally, we rescale each reward to ensure balanced contribution using weights $w_f = 1, w_a = 5, w_q = 0.4, w_c = 0.5$, demonstrated in Eqn. 4. Details for hyperparameter settings are illustrated in Appendix E.5.

$$\sigma = w_f \sigma_f + w_a \sigma_a + w_q \sigma_q + w_c \sigma_c, \tag{4}$$

### 3.2.3 MODEL TRAINING

**Group Relative Policy Optimization (GRPO).** Following Guo et al. (2025), we employ GRPO algorithm with the proposed rewards to steer the model's exploration in reasoning and retrieval. Specifically, for each training instance, we sample a group of rollouts $\{r_i\}_{i=1}^G$ from the policy model $\pi_{\theta_{\text{old}}}$ and acquire the reward $\sigma_i$. The advantage $\hat{A}_{i,t}$ is estimated as the standard score of the group rewards. GRPO objective is calculated as shown in Eqn. 5:

$$\mathcal{J}_{\text{GRPO}}(\theta) = \mathbb{E}_{x, \{r_{i,s}\}_{i=1,s=1}^{G,S} \sim \pi_{\theta_{\text{old}}}(\cdot|x,h,k)} \left\{ \frac{1}{GS|r_{i,s}|} \sum_{i=1}^{G} \sum_{s=1}^{S} \sum_{t=1}^{|r_{i,s}|} \min \left[ \rho_{i,s,t} \hat{A}_{i,s,t}, \text{clip}(\rho_{i,s,t}, 1 - \epsilon, \right. \right.$$
$$\left. \left. 1 + \epsilon) \hat{A}_{i,s,t} \right] - \beta \mathbb{D}_{\text{KL}}[\pi_\theta(\cdot|x, r_{i,<s,<t}, h_{i,<s}, k_{i,<s}) \| \pi_{\theta_{\text{ref}}}(\cdot|x, r_{i,<s,<t}, h_{i,<s}, k_{i,<s})] \right\}, \tag{5}$$

where $\rho_{i,t} = \frac{\pi_\theta(r_{i,s,t}|x, r_{i,<s,<t}, h_i, k_i)}{\pi_{\theta_{old}}(r_{i,t}|x, r_{i,<t}, h_i, k_i)}$, $h_i, k_i$ are the reasoning history and retrieved knowledge for rollout $i$. We adopt the Token-level Policy Gradient Loss (Yu et al., 2025b) to mitigate length bias, and tokens from the retrieved knowledge are masked during loss calculation.

**Two-Stage Reinforcement Learning Training.** Motivated by the empirical evidence that multimodal medical reasoning requires first establishing foundational reasoning skills before integrating additional visual information (Peng et al., 2025), we utilize a two-stage Reinforcement Learning training strategy. In the first stage, we train the model with text-only question-answer pairs sampled from MedQA-USMLE (Jin et al., 2021) and MedMCQA (Pal et al., 2022). We apply accuracy and format rewards to instill the model's fundamental medical reasoning-with-retrieval capabilities. In the second stage, we extend to multimodal training with our constructed data. All four reward types are incorporated to further encourage effective knowledge retrieval during reasoning. This two-stage training scheme allows the model to first leverage its medical reasoning-with-retrieval competencies and gradually adapt to medical imaging.

### 3.3 Confidence-Driven Test-Time Computation Scaling via Image Re-retrieval

With our proposed confidence-based reward modeling (Eqn. 3), the model is expected to output high confidence response through effective knowledge retrieval. When the model's final decision confidence remains low despite retrieval actions, it implies insufficient information in the knowledge base for the given case. This mirrors real clinical practice where clinicians find guidelines inadequate for confident decision-making. Under this circumstance, they also consult multimodal records from similar patient cases. Motivated by this, our model unlocks the capability of **C**onfidence-**D**riven test-time computation scaling via **I**mage **R**e-retrieval (**CDIR**) to enhance the diagnostic accuracy. Specifically, Eqn. 3 employs answer probability to estimate output confidence, which correlates with response correctness. We first extract the confidence $\eta$ of the generated answer token $\hat{y}_{\tau+1}^{answer}$ using Eqn. 6:

$$\eta = \pi_\theta(\hat{y}_{\tau+1}^{answer} \mid x, \hat{y}_{\leq\tau}, h, k), \tag{6}$$

where $\hat{y}$ is the output before image re-retrieval and $\hat{y}_{\tau+1}^{answer}$ denotes the probability of the predicted answer option at token position $\tau + 1$ similar to the position at §3.2.2. When the confidence score $\eta$ is below the threshold $\lambda$, i.e., $\eta < \lambda$, the model is more likely to yield an incorrect conclusion, indicating the need for multi-faceted knowledge to rectify the response. Hence, we enable the model to re-retrieve analogous cases by computing image feature similarity between the input image and candidate images from the multimodal database $\mathcal{D}$ (See Appendix C.3 for details), where each image is paired with a corresponding clinical description. The retrieved image-caption pairs are then integrated into existing contexts to re-generate the response as in Eqn. 7.

$$y \sim \pi_\theta(\cdot \mid i, t, h, k'), \text{ where } k' = \begin{cases} k, & \eta \geq \lambda \\ k \cup k_{sim}(i, \mathcal{D}), & \eta < \lambda \end{cases} \tag{7}$$

where $k_{sim}(i, \mathcal{D})$ denotes the retrieved set of similar image-caption pairs from multimodal corpus $\mathcal{D}$ based on similarity with the input image $i$. Empirically, we set $\lambda = 0.8$ in the experiment.

## 4 Experiments

### 4.1 Experimental Settings

**Public Benchmarks.** We evaluate our model on three multimodal medical benchmarks, MedXpertQA-MM (Zuo et al., 2025), MMMU-H&M (Yue et al., 2024a), and MMMU-Pro-H&M (Yue et al., 2024b). MedXpertQA-MM is a challenging benchmark collected from medical exams and textbooks. MMMU-H&M and MMMU-Pro-H&M are subsets of Health and Medicine domain derived from MMMU and MMMU-Pro benchmarks. These benchmarks target the medical MLLMs' knowledge understanding and complex reasoning abilities.

**Specialized Domain Benchmark.** To further demonstrate our model's reasoning-with-retrieval ability to generalize to unfamiliar domains, we propose a multimodal EchoCardiography Benchmark, ECBench, collected from clinical practice books. The benchmark contains 824 questions of echocardiogram interpretation in clinical scenarios. See Appendix B for curation details. We

Table 1: Performance on public multimodal medical benchmarks assessing medical MLLMs' understanding and reasoning abilities, and a curated benchmark for Echocardiology, ECBench, evaluating generalizability to unfamiliar specialty. The values marked in gray indicate the results reproduced with the officially released checkpoint. Accuracy is used as the evaluation metric. "-" denotes the checkpoint is not available for testing.

| Model | Parameter | MedXpertQA-MM | | | MMMU-H&M | MMMU-Pro-H&M | | Quilt-VQA | ECBench |
|---|---|---|---|---|---|---|---|---|---|
| | | Reasoning | Understanding | Overall | | 4 options | 10 options | | |
| *Agentic Search MLLM* | | | | | | | | | |
| Visual-ARFT (Liu et al., 2025c) | 7B | 21.0 | 22.2 | 22.0 | 58.6 | 47.6 | 30.8 | 48.4 | 41.0 |
| MMSearch-R1 (Wu et al., 2025a) | 7B | 22.3 | 23.3 | 22.6 | 56.6 | 43.4 | 26.9 | 42.9 | 41.1 |
| *Generalist Medical MLLM* | | | | | | | | | |
| MedRegA (Wang et al., 2024) | 40B | 23.1 | 28.3 | 24.6 | 47.6 | 43.4 | 25.2 | 65.9 | 37.6 |
| HuatuoGPT-Vision (Chen et al., 2024c) | 34B | 20.2 | 26.4 | 21.9 | 54.4 | 42.0 | 31.5 | 59.5 | 43.1 |
| MedGemma (Sellergren et al., 2025) | 4B | - | - | 24.4 | 47.3 | 43.7 | 32.9 | 65.6 | 41.5 |
| Lingshu (Xu et al., 2025b) | 7B | - | - | 26.7 | 54.0 | 50.0 | 37.1 | 56.9 | 44.6 |
| *Reasoning Medical MLLM* | | | | | | | | | |
| MedVLM-R1 (Pan et al., 2025) | 2B | 20.3 | 19.5 | 20.1 | 43.5 | 28.3 | 18.5 | 56.9 | 26.2 |
| Med-R1 (Lai et al., 2025) | 2B | 21.8 | 20.8 | 21.5 | 42.7 | 33.9 | 23.8 | 56.0 | 36.7 |
| GMAI-VL-R1 (Su et al., 2025) | 7B | - | - | 23.8 | 57.3 | - | 34.0 | - | - |
| XReasoner-Med (Liu et al., 2025a) | 7B | - | - | 25.9 | 63.5 | - | 40.0 | - | - |
| MedE² (Mu et al., 2025) | 7B | 25.8 | 28.5 | 26.5 | 66.0 | - | 38.8 | - | - |
| MedCCO (Rui et al., 2025) | 7B | 23.2 | 23.6 | 23.3 | 59.5 | - | - | - | - |
| Chiron-o1 (Sun et al., 2025a) | 8B | 23.3 | 25.1 | 23.8 | 54.6 | 36.7 | 24.5 | 58.0 | 36.9 |
| **MED-RWR (Ours)** | 7B | 26.2 | 29.6 | 27.2 | 65.5 | 52.5 | 43.7 | 67.9 | 51.1 |
| **MED-RWR+CDIR (Ours)** | 7B | **26.6** | **29.7** | **27.5** | **66.2** | **52.8** | **44.1** | **69.7** | **51.9** |

Table 2: Ablation Studies for Training Stages. "Text-only" denotes the first training stage with text-only data, and "Multimodal" is the second stage extending to multimodal data. To ensure fair comparison, only accuracy and format rewards are applied for all implementations.

Table 3: Ablation Studies for Reward Design. "Query" indicates the Query Semantic Reward, and "Conf" denotes the Confidence Gain Reward. Accuracy and format rewards are the default reward configurations utilized in all the experiments.

| Training Stage | | MedXpertQA-MM | MMMU-H&M | MMMU-Pro-H&M | ECBench |
|---|---|---|---|---|---|
| Text-only | Multimodal | | | | |
| ✗ | ✗ | 22.4 | 56.5 | 31.8 | 40.4 |
| ✓ | ✗ | 22.1 | 61.3 | 34.3 | 42.2 |
| ✗ | ✓ | 24.7 | 59.3 | 36.3 | 43.0 |
| ✓ | ✓ | 25.1 | 60.0 | 37.1 | 45.3 |

| Reward Design | | MedXpertQA-MM | MMMU-H&M | MMMU-Pro-H&M | ECBench |
|---|---|---|---|---|---|
| Query | Conf | | | | |
| ✗ | ✗ | 25.1 | 60.0 | 37.1 | 45.3 |
| ✓ | ✗ | 25.4 | 62.8 | 38.8 | 47.2 |
| ✗ | ✓ | 26.2 | 62.0 | 37.1 | 45.9 |
| ✓ | ✓ | **27.2** | **65.5** | **43.7** | **51.1** |

also conducted experiments on the closed set of Quilt-VQA (Seyfioglu et al., 2023), a public VQA benchmark specific for pathology domain.

**Baseline Methods.** We conduct extensive comparisons against three categories of models: (1) Agentic Search MLLMs equipped with agentic abilities to dynamically search external knowledge sources for enhanced reasoning capabilities. (2) Generalist Medical MLLM capable of perceiving diverse medical modalities and resolving versatile tasks; (3) Reasoning Medical MLLMs optimized for complex reasoning.

**Implementation Settings.** All experiments are conducted on 8 H800 GPUs using SWIFT (Zhao et al., 2025b) as the training framework. For GRPO training, we perform full-parameter fine-tuning with a learning rate of $1e^{-6}$ and $\beta$ set as $1e^{-3}$, optimized using DeepSpeed with Zero-2. Our model is finetuned from QwenVL-2.5-7B (Bai et al., 2025). More details can be found in Appendix D.

## 4.2 MAIN RESULTS

We report accuracy as the evaluation metric across all experiments. As shown in Table 1, our model achieves the highest accuracy on three complex reasoning benchmarks, showing competitive performance to larger general-purpose models despite significantly fewer parameters. MED-RWR surpasses the State-of-the-Art reasoning medical MLLM by 1% (27.5% *v.s.* 26.5%) on MedXpertQA-MM and 4.1% (44.1% *v.s.* 40.0%) on MMMU-Pro-H&M, demonstrating the efficacy of external knowledge retrieval beyond solely reasoning with internal knowledge. In addition to public benchmarks, we also perform experiments on the specialized benchmark for echocardiography, ECBench. Echocardiography is particularly challenging due to data scarcity and the high demand for clinical expertise, comprising less than 2% of both the public training corpus and ours (see Figure 4). De-

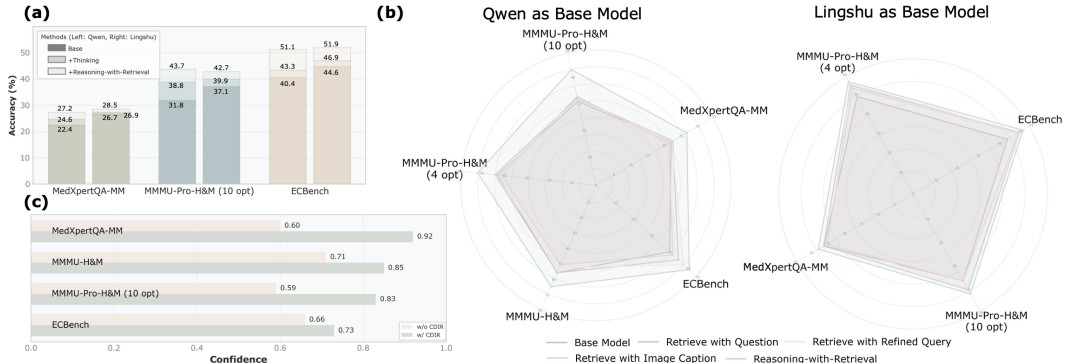

Figure 3: (a) Ablation Study of the Reasoning-with-Retrieval Capability: comparing the vanilla model, the reasoning model, and the reasoning-with-retrieval model on two architectures, QwenVL-2.5-7B (left) and Lingshu-7B (right). (b) Comparison with Training-Free RAG methods, which apply the original question, MLLM-reformulated question and MLLM-generated caption as the query, respectively. Qwen2.5-VL-7B (left) and Lingshu-7B (right) are the base models. (c) Confidence Gain after Confidence-Driven Image Re-retrieval.

spite not being specifically trained on domain-specific echocardiographic data, our model achieves an improved accuracy of 51.9%. On pathology-specific dataset Quilt-VQA, our model also achieves strong performance without any further finetuning by only proactively retrieving additional information from the domain-specific external knowledge base. This highlights the generalizability of our model to specialized medical domains by actively incorporating the external domain-specific knowledge during reasoning. Compared to Agentic Search MLLMs, our model's superior performance can be attributed to its ability to formulate more effective and context-aware search queries tailored to medical scenarios.

## 4.3 ABLATION STUDY

### 4.3.1 TRAINING DESIGN ANALYSIS

**Two-Stage Training.** We validate the necessity of two-stage training by comparing it with text-only training and direct multimodal training strategies. Table 2 demonstrate that both alternatives can improve the reasoning-with-retrieval capability and enhance model performance to some extent. However, employing RL training directly with multimodal data yields substantially smaller benefits than pretraining with text-only data first. We attribute this to the potential of text-only data to unleash the language model's reasoning ability, which manifests as more comprehensive explanations after training. In contrast, direct multimodal training requires the model to simultaneously learn multi-modal alignment and reasoning-with-retrieval abilities, which may amplify complexity and hinder the development of analytical skills.

**Reward Design.** To evaluate the impact of our proposed rewards for retrieval ability incentivization, we incrementally add them into the base model trained on the basic format and accuracy rewards. Table 3 show that both rewards can improve the performance, as evidenced by the increased accuracy. This suggests that the query semantic reward drives the model to actively seek relevant documents aligned with multimodal inputs, while the confidence gain reward ensures that the retrieved knowledge genuinely benefits final decision making.

**Reasoning-with-Retrieval Capability.** To assess the effectiveness of the reasoning-with-retrieval framework, we conduct a comparative analysis against two baselines: the base model and the reasoning model trained with reinforcement learning, which only promotes reasoning ability but excludes query and retrieval actions. We apply two base architectures for ablation, QwenVL-2.5-7B (Bai et al., 2025) and Lingshu-7B (Xu et al., 2025b). Figure 3 (a) demonstrates that incorporating reasoning-with-retrieval capabilities improves performance regardless of whether the base model have undergone medical pretraining. This improvement stems from the model's enhanced ability to actively seek relevant information from external knowledge sources during the reasoning process.

### 4.3.2 Knowledge Retrieval Analysis

**Comparison against Training-Free RAG Method with Different Query Formulation.** To demonstrate how our proposed RL training strategy enhances medical MLLM's interaction with the external medical knowledge base, we compare MED-RwR with training-free RAG method, where the query is formulated from the input question to retrieve information during inference. We implement three query formulation approaches: (1) Retrieve with Question: original question as the query; (2) Retrieve with Refined Query: MLLM-refined question as the query; (3) Retrieve with Image Caption: MLLM-generated image caption as the query. Results are shown in Figure 3 (b), with Qwen2.5-VL-7B (left) and Lingshu-7B (right) utilized as the base model. While standard RAG methods achieve moderate improvements over the base model, our framework demonstrates more effective engagement with the medical knowledge base. This is due to the tailored reward design, which elicits targeted retrieval and integration of essential knowledge.

**Confidence-Driven Image Re-retrieval.** Confidence-Driven Image Re-retrieval (CDIR) selectively triggers image re-retrieval when the model's initial confidence falls below a predefined threshold of 0.8. As demonstrated in Table 1, CDIR provides consistent performance improvements across all the benchmarks while maintaining a lightweight inference overhead. Moreover, Figure 3 presents the average confidence of the uncertain cases. CDIR substantially improves model confidence by 0.12 to 0.21 across datasets, enabling more accurate and more reliable decision-making. Supplementary experiments and case studies are included in Appendix E and F.

## 5 Conclusion

In this paper, we introduce MED-RwR, the first comprehensive Multimodal Medical Reasoning-with-Retrieval framework. We propose a two-stage reinforcement learning strategy guided by rewards, facilitating proactive retrieval of knowledge aligned with both visual findings and textual contexts. To augment potentially insufficient retrieval, we design a confidence-driven image re-retrieval mechanism for test-time scaling when model uncertainty is detected. Experiments show that MED-RwR achieves significant improvements on public benchmarks, demonstrating the crucial role of self-initiated external knowledge retrieval in complex medical reasoning. To evaluate generalizability, we evaluate on a specialized echocardiography benchmark, ECBench, which highlights the advantage of our model in leveraging external knowledge to adapt to unfamiliar domains.

### Ethics Statement

This work adheres to the ICLR Code of Ethics. All datasets used were acquired in compliance with their respective licenses and usage terms, ensuring adherence to privacy and ethical requirements. No personally identifiable data was utilized.

### Reproducibility Statement

We will release all source code and datasets in a publicly accessible repository after acceptance to ensure reproducibility. The paper includes a thorough description of the experimental setup, including model architectures, training methodologies and hyperparameter settings.

### Usage of Large Language Models

Large Language Models (LLMs) were used to polish languages and refine structures in paper writing. Additionally, LLMs also assist in data curation and evaluation process as described in Appendix A, B and E.4.

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

# APPENDIX

## A TRAINING DATA CONSTRUCTION DETAILS

### A.1 DATASET STATISTICS

The Cluster-Reconstruct-Stratify data synthesis pipeline yields 6,500 multimodal question-answer pairs. We also verify that no instances in the training data overlap with evaluation benchmarks. Figure 4 presents the modality and body structure distribution of our constructed multimodal dataset. In statistics, our dataset covers at least 10 medical modalities of various body parts. Notably, echocardiography data comprises less than 2% of our training corpus, reflecting severe underrepresentation.

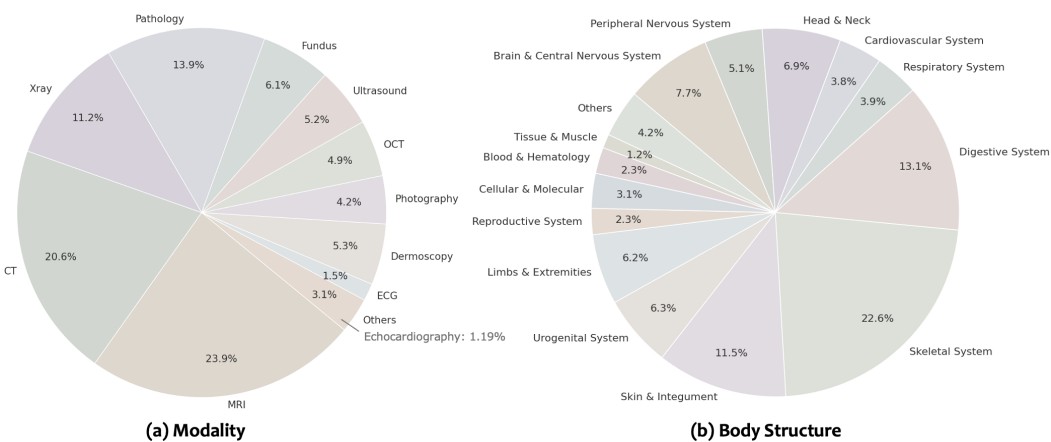

(a) Modality            (b) Body Structure

Figure 4: Modality and Body Structure Distribution of Our Constructed Multimodal Dataset.

### A.2 PROMPTS FOR DATA CONSTRUCTION

We curate training data from a public instruction tuning dataset, PubMedVision (Chen et al., 2024c), following a two-step processing with GPT-4o. For synthesizing visual question answering data, we first prompt GPT-4o to extract basic background, analysis and conclusion from the original data. The prompt is displayed in Table 4.

Table 4: Prompt for Initial Information Extraction from Existing VQA Samples.

| **Prompt Template for Extracting Information from Original Data** |
| :--- |
| You will be provided with a visual question-answer pair, which includes an image, a question, and an answer. Your task is to analyze this information and extract the clinical information (patient basic information, medical history, or laboratory results), observation, analysis and conclusion from the given contents. Format the output as JSON dictionary. |

Sequentially, we further prompt GPT-4o to formulate confusable options according to the provided descriptions, with the prompt in Table 5. In Figure 5, we present an example to illustrate the complete data synthesis process.

## B ECHOCARDIOGRAPHY BENCHMARK (ECBENCH) CONSTRUCTION DETAILS

To establish a benchmark that provides objective evaluations for echocardiography, we utilized three authoritative echocardiography practice books as the foundational sources for the benchmark: (1)

Table 5: Prompt for Training Data Construction Given Extracted Information.

---

**Prompt Template for Constructing Training Data from the Extracted Information**

---

Please construct a standard and complex medical exam question and answer based on the example. You are provided with an image alongside with the observation, analysis and conclusion. Please follow the following instruction:

1. Create a new question in a complex and standard format. If clinical information about the patient is available, such as basic information, medical history, or laboratory results, incorporate these relevant clinical details into the question to provide appropriate context for the diagnostic scenario.

2. Since the original question requires an image, the constructed question should also require observing an image to answer. Do not indicate what the image contains.

3. Include the correct answer and up to three wrong but plausible/confusing options (no more than four options total). Randomize the position of the correct answer.

4. The constructed question should not contain any additional information that is not provided and mentioned in the given contexts. Only use the existing information.

Format the output as JSON containing:

"question": The question

"options": A list of answer choices, each prefixed with a letter label (e.g., "A. ...", "B. ...").

"answer": The correct answer, indicated by letter (e.g., "B. ...").

"explanation": The explanation of the correct answer.

---

**Original VQA from PubMedVision**

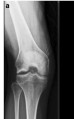 Human: What is the main finding observed in the radiographic images of the knee?
Assistant: The main finding observed in the radiographic images of the knee is a large osteochondral defect on the lateral femoral condyle of the right knee joint. This defect appears as an area of irregularity and decreased bone density within the joint surface, indicating a loss or damage to the cartilage and underlying bone in that region. This finding is consistent with the reference information provided, which states that the radiographic images showed a large osteochondral defect on the lateral femoral condyle of the right knee in a 31-year-old female patient.

**Extracted Information**

**clinical information**: patient_basic_information: age: 31, gender: female, medical_history: None, laboratory_results: None.
**observation**: The radiographic image of the knee reveals a large osteochondral defect on the lateral femoral condyle of the right knee. The defect is characterized by irregularity and decreased bone density within the joint surface.
**analysis**: The osteochondral defect suggests significant damage to the cartilage and underlying bone, likely leading to joint instability or dysfunction. This finding correlates with the clinical information provided, further supporting the diagnosis. The defect is substantial and may require advanced therapeutic intervention.
**conclusion**: The primary finding in the radiographic imaging is a large osteochondral defect on the lateral femoral condyle of the right knee. This condition is indicative of significant cartilage and bone damage, requiring further clinical evaluation and potential treatment, such as surgical repair or other interventions.

**Reformulated VQA**

**question**: A 31-year-old female presents with persistent knee pain and instability. A radiographic image of her right knee is provided. Based on the image, what is the most likely diagnosis?
**options**: A. Osteochondral defect of the lateral femoral condyle, B. Medial meniscus tear, C. Patellar dislocation, D. Osteoarthritis of the medial compartment
**answer**: A. Osteochondral defect of the lateral femoral condyle
**explanation**: The radiographic image shows a large osteochondral defect on the lateral femoral condyle of the right knee, characterized by irregularity and decreased bone density. This matches the description of cartilage and bone damage, which is the hallmark of an osteochondral defect. Other options, such as a medial meniscus tear or patellar dislocation, would not present with these specific radiographic features, and osteoarthritis typically manifests with joint space narrowing and osteophyte formation, which are not observed here.

Figure 5: Illustration of the Data Synthesis Process.

Critical Care Echocardiography: A Self-Assessment Book (Sreedharan et al., 2024), (2) Echocardiography Board Review: 600 Multiple Choice Questions with Discussion (Pai & Varadarajan, 2024), (3) Complex Cases in Echocardiography (Siegel et al., 2014). Importantly, these three practice books were not incorporated into the model development process to maintain fairness and prevent data leakage.

These books offer comprehensive coverage, including foundational knowledge of echocardiography and case studies relevant to echocardiographic diagnosis. Each question is formatted as a multiple-choice question, accompanied by its ground truth answer and an explanatory paragraph detailing the rationale. Furthermore, a significant proportion of the questions are supplemented with corresponding echocardiographic images for diagnostic observation.

Adopting the methodology outlined in Qin et al. (2025), the construction of the ECBench benchmark involves a three-stage process: (1) extracting raw content from the practice books using the state-of-the-art Mistral OCR[1], which converts content into markdown format; (2) utilizing GPT-4o to structure the markdown content into a list of dictionaries, with each entry containing the question, its answer, the rationale, and any associated multimedia content; and (3) employing GPT-4o to refine the output by removing redundant inline references. The detailed prompts used for the extraction process are presented in Tables 6 and 7.

Table 6: Prompt for Organizing Raw Markdown Content.

---

**Prompt for Organizing Raw Markdown Content**

---

You are given a paragraph containing potential four types of content: questions, images in questions, answers, and explanations for answers. This paragraph is in markdown format. Your task is to organize the raw content into an structured dict, you should return a json list, which each of the item of the json dict looks like this: 'question':'(the complete question including the background information)', 'candidates': 'the candidate', 'answer': 'the ground truth answer', 'explanation': 'the explanation to the answer', 'image': [The list of image paths that are involved in the question.]. Note the image path should be only in the one in the markdown image reference format. Note that the stem (clinical background) of the question should also be properly included in each question, if included in the paragraph. If there's no content for a specific entry, fill the entry with 'None'. Do not summarize the content. Return the list of dict only. The paragraph is: [paragraph]

---

Table 7: Prompt for Clean Up Inline References.

---

**Prompt for Clean Up Inline References**

---

You are given a question. Please clean the content by: 1. remove inline image reference (e.g., Figure/Fig. 1.1 ..., Video/Vid. 1.1. ...), 2. if there's any candidate choice, remove them. Do not make other modifications. If there's no need for cleaning, just return the original question. Return the cleaned question only. This is the question: [question]

---

## C  KNOWLEDGE BASE CONSTRUCTION DETAILS

### C.1  CONSTRUCTING KNOWLEDGE BASES FROM QUESTION EXPLANATION

To support the retrieval component of our MED-RWR framework, we construct a medical knowledge base by extracting explanatory content from available training question-answer pairs with GPT-4o. We utilize explanations from text-only datasets such as MedMCQA (Pal et al., 2022), which contain detailed explanations from real-world medical entrance exam questions. Additionally, we also extract relevant medical knowledge from the intermediate analysis generated during our multimodal training data construction process. The employed prompt can be found in Table 8. This knowledge base encompasses medical terminology illustration, analysis of symptoms and signs, disease manifestations, and treatment protocols, as exemplified in Figure 6. By utilizing knowledge derived from highly related explanations, we ensure that the model is more likely to successfully retrieve information when querying the knowledge base during reasoning, allowing us to concentrate on evoking the model's reasoning-with-retrieval ability. Through this approach, we obtain a knowledge base, **ExpKB**, which supports the retrieve-while-reasoning training. Similarly, we construct **EC-ExpKB** from the explanations in echocardiography practice collections as a specialized knowledge base supporting model generalization to the echocardiography domain.

### C.2  APPLYING MEDICAL RESOURCES AS KNOWLEDGE BASES

To assess the generalizability and practical applicability of our approach, we conducted experiments using two additional external knowledge bases derived from medical resources, PubMed[2]

---

[1]https://mistral.ai/news/mistral-ocr
[2]https://ftp.ncbi.nlm.nih.gov/pub/lu/MedCPT/pubmed_embeddings

Table 8: Prompt for Medical Knowledge Extraction.

---

**Prompt for Extracting Medical Knowledge from Explanations of Exam Questions.**

---

Extract essential medical knowledge from the following medical exam question explanation and format the output as JSON.

FILTER OUT these irrelevant elements: Answer designations, Citations and references, Formatting artifacts and special characters, Question numbers or identifiers

The JSON output should INCLUDE ONLY:

1. "medical_terminology": A list of sentences defining specialized medical terms (e.g., "Myocardial infarction is the death of heart muscle tissue due to inadequate blood supply from coronary artery occlusion.")

2. "symptoms_and_signs": A list of sentences describing what patients experience or what clinicians observe when having a specific disease or condition (e.g., "Patients with myocardial infarction typically present with severe crushing chest pain.")

3. "disease_manifestations": A list of sentences describing how the disease presents, progresses, or affects the body (e.g., "Myocardial infarction can lead to progressive heart failure over time.", "Patients with myocardial infarction may develop life-threatening arrhythmias within 24-48 hours of the initial event.")

4. "treatment_protocols": A list of sentences describing specific medical interventions and management approaches (e.g., "Emergency treatment for myocardial infarction includes immediate administration of aspirin 325mg chewed.", "Primary percutaneous coronary intervention should be performed within 90 minutes of presentation if STEMI is diagnosed.")

Please extract only the essential medical knowledge from the following explanation using the specified JSON format. If no relevant information exists for a particular category, include that key with a empty list.

---

**Original Explanation**

B. i.e. Frontoparietal sutureSkull sutures (except sphen-occipital), vomer- sphenoid rostrum schindylesisQ, inferior tibiofibular syndesmosis and dentoalveolar gomphosis (tooth socket) joints are fibrous joints. A Schindylesis is a specialized suture where a rigid bone fits into a groove on a neighbouring element eg a cleft between the alae of vomer, which receives the rostrum of the sphenoid. (i.e. vomer - sphenoidal rostrum junction)Q- Epiphyseo-diaphyseal growth plateQ, costochondral, 1st chondrosternal and sphenoocipital joints are synchondrosis/ syncendrosis or primary cailaginous joints; whereas, symphysis pubis, symphysis menti, sacrococcygeal, interveebral and manubriosternal jointsQ are symphyses or secondary cailaginous joints.Aicular cailage of most (not all) synol joints are lined by hyaline cailage Q. In some synol joints the cavity is paially or completely subdivided into two by an aicular disc, made up of fibrocailage eg. Temporomadibular jt, sternoclavicular jt, menisci of knee joint. More mobile the joint the more commonly it discloates eg shoulder jtQ. MCP Jt. is a condylar - synol joint

**Extracted Medical Knowledge**

**medical_terminology:** ["A schindylesis is a specialized suture where a rigid bone fits into a groove on a neighboring element, such as the cleft between the alae of the vomer, which receives the rostrum of the sphenoid.", "Synchondrosis, also called primary cartilaginous joints, include the epiphyseo-diaphyseal growth plate, costochondral joints, first chondrosternal joint, and spheno-occipital joint.", "Symphysis joints, also called secondary cartilaginous joints, include the symphysis pubis, symphysis menti, sacrococcygeal joints, intervertebral joints, and manubriosternal joints.", "Articular cartilage of most synovial joints is lined by hyaline cartilage.", "Synovial joints can have their cavities partially or completely subdivided by an articular disc made of fibrocartilage, such as in the temporomandibular joint, sternoclavicular joint, and menisci of the knee joint.", "MCP (metacarpophalangeal) joint is a condylar synovial joint."]

**symptoms and signs:** []

**disease manifestations:** ["More mobile joints, such as the shoulder joint, are more commonly dislocated."]

**treatment protocols:** []

Figure 6: Example of Extracting Medical Knowledge from Explanation.

and StatPearls[3]. While utilizing knowledge bases constructed from question explanations ensures a controlled environment for evoking reasoning-with-retrieval ability, it is also crucial to evaluate whether our proposed MED-RwR framework can effectively leverage real-world medical knowledge sources. For PubMed, we leverage the articles processed by Jin et al. (2023), which includes abstracts from leading medical journals across diverse specialties. For StatPearls, we systematically crawl the detailed explanations of medical concepts, which covers knowledge of diseases, medical conditions, interventions, diagnostic procedures, treatments, clinical guidelines, and etc.

## C.3 MULTIMODAL CORPUS

We utilize a multimodal corpus with image-caption pairs for Confidence-Driven Image Retrieval. For experiments on public benchmarks, we apply the multimodal database from PubMed processed

---

[3]https://www.ncbi.nlm.nih.gov/books/NBK430685

by Chen et al. (2024c), PubMedVision, which includes figures with their annotated caption sourced from PubMed papers, denoted as PubMed-MMKB. For experiments on the Echocardiology domain, we apply a multimodal echocardiography expertise database, ECED (Qin et al., 2025), which includes diverse echocardiology images with their detailed descriptions. Table 9 summarizes the size of all knowledge bases used in our experiments.

Table 9: Size of Utilized Knowledge Bases.

| Knowledge Base | Modality | Entries |
|---|---|---|
| PubMedKB | Text | 993,472 |
| StatPearlsKB | Text | 1,023,144 |
| ExpKB | Text | 58,020 |
| EC-ExpKB | Text | 1,577 |
| PubMedVision | Text & Image | 647,031 |
| ECED | Text & Image | 11,732 |

## D  TRAINING IMPLEMENTATION DETAILS

### D.1  PROMPT DESIGN

To guide the model to follow the predefined structure, we first craft a prompt template shown in Table 10. The model is allowed to conduct multiple rounds of thinking and retrieval before finally reaching a decision.

Table 10: Prompt for Evoking Reasoning-with-Retrieval

**Prompt Template for MED-RWR**

You are an experienced expert in medicine. You are given a question, an image and a list of choices. You are required to select the correct answer from the choices. First observe the image, think about the question and each choice within `<think> </think>` tags. During thinking, if needed, retrieve medical knowledge using `<query> </query>` tags. Only one query is allowed. An external agent will retrieve information and return it within `<retrieve> </retrieve>` tags. You can use the retrieved information to continue thinking and further query if more information is needed. When you can reach a conclusion, output your answer within `<answer> </answer>` tags.

The output should be in the following format:

1.  If you need more information, output `<think>...</think>` `<query>...</query>` `<retrieve>...</retrieve>`. Multiple think-query-retrieve cycles may occur.

2.  If you can directly reach a conclusion without query, output `<think>...</think>` `<answer>...</answer>`.

### D.2  RETRIEVAL IMPLEMENTATION

Our paper incorporates two types of retrieval operations: (1) text-based retrieval during the reasoning-with-retrieval process (§ D.2.1); (2) image-based retrieval applied in Confidence-Driven Image Retrieval (§ D.2.2) for test-time computational scaling.

### D.2.1  REASONING WITH RETRIEVAL

As described in § 3.2.1, during the rollout process, the model first observes the image and analyzes the textual information through a reasoning process. When the model determines that its internal knowledge may be insufficient for an accurate response, it generates a query to retrieve relevant information from an external medical knowledge database. During the retrieval process, we input the query into retriever to search for related documents from the knowledge base. For retriever, we employ the pretrained Med-CPT (Jin et al., 2023) for PubMedKB and BGE-M3 (Chen et al., 2024a) for the remaining KBs. Med-CPT is an information retrieval model specifically designed for the biomedicine domain, pretrained with query-article pairs from PubMed, making it ideal to retrieve from PubMed sources. BGE-M3 is a highly versatile embedding model that supports both

embedding and retrieval, capable to process inputs of different scales from short sentences to long documents. We divide each item into chunks of at least 100 words for efficient indexing and retrieval. The 3 most relevant documents are retrieved for each query during the retrieval process.

To ensure a fair comparison, we adjust the baseline setup for the Agentic Search MLLMs. Instead of applying their original online search environments, the baselines were configured to use the same knowledge base and retriever as our proposed framework.

### D.2.2 CONFIDENCE-DRIVEN IMAGE RETRIEVAL

During inference, we compensate the model with additional information from multi-mocal cases when the model's confidence in its answer is low, indicating insufficient knowledge obtained from text-based retrieval, as illustrated in § 3.3. To retrieve similar multimodal cases, we obtain the features of input image using BioMedCLIP (Zhang et al., 2023a), a powerful foundation model pre-trained on millions of biomedical image-text pairs. To efficiently find similar cases, we compute the cosine similarity between the features of the input image and a subset of 10,000 randomly selected images from the corpus. The most similar case with the highest similarity scores alongside its corresponding caption are retrieved. We then integrate the retrieved image feature and text tokens with the original input and history contexts from the previous output, which is subsequently used for regenerating the response. This process leverages the additional multimodal information to improve the correctness and confidence of the final output, illustrated in Algorithm 1.

---

**Algorithm 1** Confidence-Driven Image Retrieval

---

**Input:** Input Image and Question Pair $x = [i; t]$ , Multimodal corpus $\mathcal{D}$ containing image-caption pairs
**Output:** Final Decision $y$
Generate intermediate turns of think, query, retrieve $h_n, k_n \leftarrow \pi_\theta(x, h_{1:n-1}, k_{1:n-1})$
Generate output $\hat{y} \leftarrow \pi_\theta(\cdot \mid x, h_{1:n}, k_{1:n})$
Calculate answer confidence $\eta \leftarrow \pi_\theta(\hat{y}_{\tau+1}^{\text{answer}} \mid x, \hat{y}_{\leq\tau}, h_{1:n}, k_{1:n})$
**if** $\eta < 0.8$ **then**                                                   ▷ low confidence
    Conduct image retrieval $(i_{\text{sim}}, t_{\text{sim}}) \leftarrow \|_{\text{sim}}(i, \mathcal{D})$
    $y \leftarrow \pi_\theta(\cdot \mid x, h_{1:n}, k_{1:n}, i_{\text{sim}}, t_{\text{sim}})$         ▷ incorporate the retrieved multimodal case
**else**                                                                      ▷ high confidence
    Apply the confident answer $y \leftarrow \hat{y}$
**end if**

---

### D.3 TRAINING DYNAMICS

In the first stage of training, we concentrate on eliciting the model's reasoning-with-retrieval ability with text-only question-answer pairs through accuracy and format rewards. The evolution of rewards and completion length over training steps are shown in Figure 7.

In the second stage of training, all four reward types proposed are incorporated to enhance the effectiveness of query formulation and relevance of retrieved knowledge. Figure 8 demonstrates the reward and response length change across training iterations.

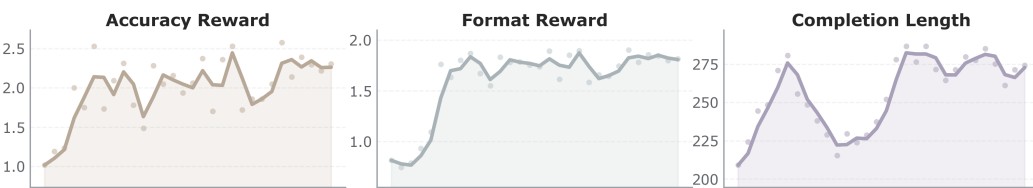

Figure 7: Rewards and Response Length across Training Iterations of the First Stage. The rewards are plotted every five steps.

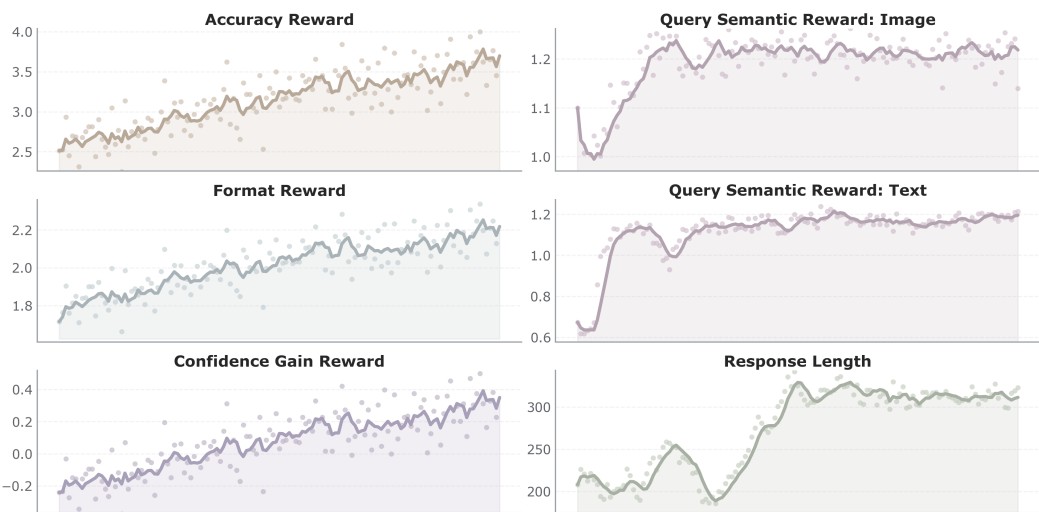

Figure 8: Rewards and Response Length across Training Iterations of the Second Stage. The rewards are plotted every five steps.

## E  SUPPLEMENTARY EXPERIMENTS

### E.1  COMPARISON WITH REASONING DATA SUPERVISED FINE-TUNING

To validate the effectiveness of our two-stage training strategy, we compared with employing text-only supervised fine-tuning (SFT) instead of GRPO training in the first stage, while maintaining multimodal GRPO training in the second stage. For the SFT data, since our text-only GRPO training data does not contain explicit reasoning chains necessary for supervised fine-tuning, we employed two publicly released medical reasoning datasets derived from the same medical QA sources,he MedQA-USMLE (Jin et al., 2021) and MedMCQA (Pal et al., 2022), as we used for GRPO training. As shown in Table 11, we attribute this performance drop to the model's tendency to overfit reasoning templates during SFT training without adequately understanding the inherent medical knowledge, which appears to constrain the model's ability to dynamically retrieve and synthesize relevant information. Future directions could explore developing more adaptive approaches to cold-start data construction in the reasoning-with-retrieval scenario.

Table 11: Comparison with Applying Supervised Fine-tuning in the First Stage to Warm up Reasoning.

| First Stage | Second Stage | MedXpertQA-MM | MMMU-H&M | MMMU-Pro-H&M | ECBench |
|---|---|---|---|---|---|
| Med-o1 SFT | Multimodal GRPO | 24.1 | 55.3 | 33.6 | 39.7 |
| MedReason SFT | Multimodal GRPO | 26.1 | 52.0 | 36.7 | 39.8 |
| Text-only GRPO | Multimodal GRPO | 27.2 | 65.5 | 43.7 | 51.1 |

### E.2  INTEGRATING DIFFERENT KNOWLEDGE BASES

To assess the generalizability and practical applicability of our approach toward different Knowledge Bases (KBs), we conduct experiments with different KBs described in C.1 and present the results in Table 12. The results demonstrate that the model achieves the best performance with our constructed KB derived from exam explanations on general medical QA tasks, while showing optimal results in echocardiography domain with the specialized echocardiography KB. This is probably because our constructed KBs contain fewer but more targeted entries as shown in the KB statistics (Table 9). This enables the retriever to more effectively identify relevant contents for each query. In contrast, larger-scale KBs, while containing more comprehensive knowledge, present greater challenges for the retrieval systems. This suggests that for knowledge bases with substantial scale and diversity, further developing a more sophisticated retrieval mechanism becomes essential.

Table 12: Results of Applying External Knowledge from Different KBs.

|  | MedXpertQA-MM | MMMU-H&M | MMMU-Pro-H&M | ECBench |
|---|---|---|---|---|
| PubMedKB | 26.4 | 61.3 | 41.3 | 49.0 |
| StatpearlsKB | 26.6 | 61.4 | 42.0 | 49.3 |
| ExpKB | **27.2** | **65.5** | **43.7** | 48.3 |
| EC-ExpKB | 25.4 | 62.1 | 40.2 | **51.1** |

### E.3 Experiments on Traditional Medical VQA Datasets

We also report the accuracy on traditional medical VQA datasets, SLAKE (Liu et al., 2021), VQA-RAD (Lau et al., 2018), PathVQA (He et al., 2020), PMC-VQA (Zhang et al., 2023b), and Omni-iMedVQA (Hu et al., 2024), in Table 13. Note that *our model is not trained with any data from the training split of these target datasets*. Compared with the base model QwenVL-2.5, the performance gains achieved by our approach are consistent but not substantial. This can possibly be attributed to the difference in dataset complexity and the requirements of reasoning abilities. These datasets contain large amounts of questions that can be directly inferred from images without requiring complicated reasoning or external knowledge, and thus do not necessitate the reasoning-with-retrieval capability our model enhances.

Table 13: Performance on Traditional Medical VQA Datasets.

| Model | SLAKE | VQA-RAD | PathVQA | PMC-VQA | OmniMedVQA |
|---|---|---|---|---|---|
| QwenVL-2.5 | 74.0 | 73.1 | 64.8 | 53.0 | 63.9 |
| MED-RWR | 75.5 | 75.4 | 69.9 | 53.5 | 73.1 |

### E.4 Evaluating Reasoning Path with the LLM-as-a-Judge Strategy

We further apply an LLM-as-a-Judge strategy to evaluate the quality of the reasoning path. Specifically, we apply Qwen3-32B (Yang et al., 2025a) to assess the generated reasoning path on ECBench from four aspects: (1) Coherence: assesses if the generated reasoning is coherent, grammatically correct, and fluent; (2) Factuality: assess if the reasoning is grounded on correct and verifiable clinical knowledge; (3) Relevance: assess if the reasoning directly addresses the problem posed in the question and logically supports the given answer; (4) Conciseness: assess if the reasoning avoid redundant steps and irrelevant details. A score between 0 and 5 is assigned to each aspect. Results are shown in Table 14.

Table 14: Reasoning Path Evaluation with LLM-as-a-Judge Strategy.

| Model | Coherence | Factuality | Relevance | Conciseness | Average |
|---|---|---|---|---|---|
| MedVLM-R1 (Pan et al., 2025) | 3.2 | 1.7 | 2.4 | 3.4 | 2.7 |
| Med-R1 (Lai et al., 2025) | 3.7 | 2.1 | 2.4 | 4.0 | 3.1 |
| Chiron-o1 (Sun et al., 2025a) | 4.2 | 2.8 | 3.5 | 4.0 | 3.6 |
| MED-RWR (Ours) | **4.4** | **3.1** | **3.8** | **4.4** | **3.9** |

### E.5 Sensitivity Analysis of Reward Weights

In reward design, we introduce hyperparameters $w_f = 1, w_a = 5, w_q = 0.4, w_c = 0.5$ to control the reward weights for Format Reward, Accuracy Reward, Query Semantic Reward, and Confidence Gain Reward, respectively. Notably, we define these hyperparameters based on the following principles: (1) prioritizing the dominant task objective, which is the accuracy reward guided by $w_a$, and (2) balancing the scale of other supporting rewards with $w_f, w_q, w_c$ to ensure stable training. Figure 8 shows that our chosen weights lead to a stable training process, where all reward components are rescaled into a similar and reasonable range. Specifically, we intentionally use a smaller weight for Confidence Gain Reward to prevent the model from overemphasizing confidence scores at the expense of final decision accuracy and retrieved information quality. The results in Table 15 show

that our configuration achieves the best performance compared to other weight settings, demonstrating that prioritizing accuracy while maintaining balanced reward scales is crucial for stable and effective training.

Table 15: Sensitivity Analysis of Reward Weights

| Setting | $w_f$ | $w_a$ | $w_q$ | $w_c$ | MedXpertQA | MMMU | MMMUPro | ECBench |
|---------|-------|-------|-------|-------|------------|------|---------|---------|
| All Same Weights | 1 | 1 | 1 | 1 | 23.4 | 57.6 | 37.4 | 43.1 |
| Low Accuracy Weight | 1 | 1 | 0.4 | 0.5 | 25.1 | 64.8 | 40.9 | 47.1 |
| High Query Semantic Weight | 1 | 5 | 1 | 0.5 | 25.5 | **66.0** | 41.9 | 49.6 |
| High Confidence Gain Weight | 1 | 5 | 0.4 | 1 | 26.2 | 62.8 | 41.3 | 50.0 |
| Ours | 1 | 5 | 0.4 | 0.5 | **27.2** | 65.5 | **43.7** | **51.1** |

### E.6 COMPARING WITH MULTIMODAL RAG BASELINES IN MEDICAL DOMAIN

Current works like MMed-RAG (Xia et al., 2025b) and RULE (Xia et al., 2024) have explored to enhance the medical MLLM with multimodal RAG methods by applying modality-specific retrievers to retrieve related medical reports based on images. We compared our methods with these medical-domain RAG methods in the same setting as shown in Table 16. To maintain a fair comparison, we finetune our model on the same training set and evaluate on the same test dataset as Xia et al. (2025b). Results of MMed-RAG and RULE are referenced from their original paper. The results demonstrated that on data domain where image retriever and medical reports are available, our proposed reasoning-with-retrieval paradigm still has better performance over strong multimodal RAG baselines for the medical domain.

Table 16: Comparison with Multimodal RAG Baselines in Medical Domain

| Model | IU-Xray | MIMIC-CXR | Quilt-1M | PMC-OA Pathology |
|-------|---------|-----------|----------|------------------|
| RULE (Xia et al., 2024) | 87.84 | **83.92** | 68.97 | 61.41 |
| MMed-RAG (Xia et al., 2025b) | 89.54 | 83.57 | 72.95 | 64.54 |
| MED-RWR | **92.81** | **83.92** | **75.74** | **67.35** |

### E.7 ANALYSIS ON SCALABILITY OF CONFIDENCE-DRIVEN IMAGE RE-RETRIEVAL

We further conducted a set of experiments to illustrate the scalability of Confidence-Driven Image Re-retrieval (CDIR) mechanism.

**Increasing the Scale of Retrieved Cases.** We have conducted ablation studies increasing the number of retrieved cases from 0 to 5. As shown in the Figure 9 (a), the performance initially increases then decreases as the number of retrieved cases grows, indicating that retrieving more cases not necessarily provides consistent performance improvement. We attribute this to the diminishing similarity of increased samples. Retrieving more samples could expose the model to less similar information, which would potentially confuse the model with noisy information.

**Increasing the Scale of Retrieval Corpus.** We further conduct experiment expanding the retrieval pool from 10,000 to 50,000 images, where we observe a consistent performance improvement. This demonstrates that the effectiveness of CDIR mechanism scales up with corpus size. A larger corpus increases the likelihood of finding highly similar cases, thus providing more relevant information to the model. Thus, CDIR can be further strengthened as the size of retrieval corpus grows.

## F CASE STUDIES

### F.1 CASE STUDIES FOR REASONING-WITH-RETRIEVAL PROCESS

We demonstrate several case studies of the rollout process generated with our proposed reasoning-with-retrieval framework MED-RWR, revealing how the retrieval mechanism could evolve with clinical reasoning. Figure 10 provides an example of *observation-driven retrieval* when model encounter unfamiliar visual observations in the medical scan, such as the anatomical structure "splenic

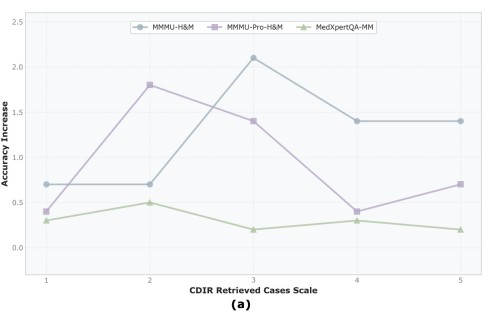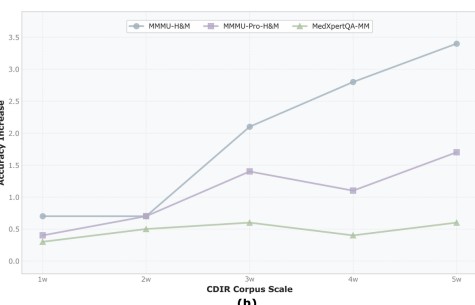

**(a)**    **(b)**

Figure 9: Scalability of Confidence-Driven Image Re-retrieval (CDIR): performance variation of scaling up (a) size of retrieved case and (b) multimodal corpus size. The y-axis denotes the percentage increase in accuracy

sinusoids". This triggers the model to retrieve relevant knowledge, including definitions and physiological functions, which may potentially help interpret the image and reach the final conclusion. *Hypothesis-driven retrieval* emerges as shown in Figure 11 when the model can relate visual findings with a specific clinical symptom to formulate a preliminary diagnosis, but requires additional information on treatment pathways, prognostic assessments, and so forth. When the model possesses sufficient internal knowledge to reach clinical conclusions as an example in Figure 12, *confirmation-driven retrieval* is also helpful to ensure adherence to clinical protocols and ground reasoning in authoritative guidelines.

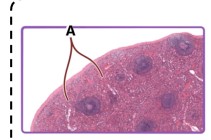

In a microscopic examination of splenic tissue, a researcher is analyzing the cellular composition and structural organization of the spleen. Upon examining a biopsy specimen under the microscope, as depicted in the image, what is the most accurate description of the region labeled A?
A. The marginal center where antigen presentation occurs
B. It contains a group of mature T-lymphocytes that surround a central arteriole and is part of the adaptive immune system.
C. It contains dense connective tissue and surrounds the spleen
D. It is a highly vascularized area containing sinusoids and is responsible for removing old red blood cells.
E. Contains the germinal centers

<think> The region labeled A in the image corresponds to the **splenic sinusoids**, which are highly vascularized areas within the splenic cords. </think>
<query> splenic sinusoids function histology </query>
<retrieve> (1) The spleen's function is divided between its two main components, the red and white pulp. The red pulp acts as a blood filter, consisting of splenic cords and sinusoids that **test the flexibility of red blood cells, removing and destroying old or damaged ones**. It also serves as a reservoir for platelets. The white pulp functions as a large lymphoid organ, constantly **monitoring the blood for pathogens**. It is critical for initiating immune responses, especially against encapsulated bacteria like Streptococcus pneumoniae. (2) The sphenoid sinuses are air-filled cavities located centrally within the sphenoid bone, whose importance is defined by the critical structures surrounding them. Superior to the sinus is the sella turcica, which houses the pituitary gland, and the optic chiasm. Laterally, the sinus is bordered by the cavernous sinuses, which contain the internal carotid arteries and cranial nerves III, IV, V, and VI, all essential for eye movement and facial sensation. This proximity makes the sinus a key surgical pathway to the pituitary and a site of potentially dangerous infections.(3) Liver sinusoids are specialized, low-pressure vascular channels that receive a dual blood supply from the portal vein and hepatic artery. Their walls are lined by a discontinuous, fenestrated endothelium that lacks a basement membrane, allowing blood plasma to have direct contact with hepatocytes for metabolic processing. Within the sinusoidal lining are Kupffer cells, which are resident macrophages that play a crucial role in clearing bacteria and endotoxins from the portal blood coming from the intestines. This unique structure is essential for the liver's detoxification and synthetic functions. </retrieve>
<think>Based on the retrieved information, these sinusoids are responsible for **removing old red blood cells through phagocytosis by macrophages**. This process is crucial for maintaining the circulatory system's efficiency by recycling hemoglobin and preventing the accumulation of senescent erythrocytes. The most accurate description of the region labeled A is D. It is a **highly vascularized area containing sinusoids and is responsible for removing old red blood cells.** </think>
<answer> D. It is a highly vascularized area containing sinusoids and is responsible for removing old red blood cells. </answer>

Figure 10: Example of MED-RWR's Rollout Process: Observation-Driven Retrieval.

## F.2 CASE STUDIES FOR CONFIDENCE-DRIVEN IMAGE RE-RETRIEVAL

To illustrate the efficacy of the Confidence-Driven Image Re-retrieval (CDIR) mechanism, we present a case study in Figure 13 and Figure 14, which demonstrate the rollout process with and without CDIR. Initially, while the model formulated an appropriate and clinically relevant query "knee meniscal tear arthritis progression" based on the input image and clinical context, the knowledge retrieval process failed to return helpful information from the knowledge base. The retrieval ineffectiveness resulted in incorrect decision-making, as it failed to help the model differentiate the clinical significance of an ACL injury from a meniscal tear in this context. This is potentially reflected by the model's low confidence, which fell below the predefined threshold $\lambda = 0.8$, so CDIR mechanism is triggered. The model then performed a re-retrieval by searching the multimodal database for

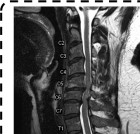 A 63-year-old woman presents with a broad-based shuffling gait, reduced manual dexterity, and exaggerated deep tendon reflexes in the lower extremities. A T2-weighted MRI scan (Figure A) is provided. What is the most appropriate treatment for the patient?
A. Immobilization in a halo orthosis for 6 weeks followed by gradual ROM exercises
B. C4 to C7 cervical laminectomy with fusion
C. Multilevel anterior cervical decompression with fusion and stabilization
D. C4 to C7 laminoplasty with plate fixation
E. C4 to C7 cervical laminectomy

<think> The MRI shows a large central herniated nucleus pulposus at C5-C6 with significant spinal cord compression. The patient's symptoms suggest a cervical myelopathy. </think>
<query> surgical management cervical myelopathy </query>
<retrieve> (1) For most patients with isolated back discomfort due to DISH, primary treatment options include activity modification, physical therapy, bracing, nonsteroidal anti-inflammatory drugs (NSAIDs), and bisphosphonates. These interventions aim to alleviate symptoms, improve mobility, and prevent further complications associated with the condition.. **Surgical decompression and stabilization may be indicated for specific sequelae of DISH, such as fractures, cervical myelopathy, lumbar stenosis, neurologic deficits, infections, or painful deformities.** Early intervention is essential to prevent symptom progression and improve overall outcomes. (2) Degenerative cervical myelopathy is a spinal cord dysfunction that can lead to numbness, pain, imbalance, sensory loss, hyperreflexia, and urinary incontinence. Estimates are that nearly 1.6 per 100,000 have had surgical treatment for cervical myelopathy. During a standard neurological exam, the Hoffman sign is common, and a positive sign can aid in the diagnosis. (3) The gold standard for diagnosing cervical myelopathy is magnetic resonance imaging (MRI). The Hoffmann sign is not always indicative of pathological cord compression. But when it is present with corresponding MRI images, it was shown that if the compression of the cervical cord was higher in the canal, it correlated with a higher incidence of a positive Hoffman sign. </retrieve>
<think> For such cases, surgical intervention is typically necessary to decompress the spinal cord and relieve symptoms. Given the extent of the herniation and the need for stabilization, a multilevel anterior cervical decompression with fusion and stabilization would be the most appropriate treatment option. The most appropriate treatment for the patient is C. Multilevel anterior cervical decompression with fusion and stabilization. </think>
<answer> C. Multilevel anterior cervical decompression with fusion and stabilization </answer>

Figure 11: Example of MED-RWR's Rollout Process: Hypothesis-Driven Retrieval.

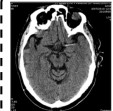 A noncontrast CT scan of the head is performed immediately, and the results are shown. The patient's vital signs include a temperature of 37.1°C (98.8°F), a pulse of 101/min, and a blood pressure of 174/102 mm Hg. What is the most appropriate next step in this patient's management?
A. Surgical clipping
B. Intravenous labetalol therapy
C. Decompressive surgery
D. Intravenous

<think> The image shows a large area of low-density lesion in the right hemisphere of the brain, suggestive of a large vessel occlusion, which is consistent with an acute ischemic stroke. The presence of a large infarct without significant mass effect or midline shift suggests that the patient might be a candidate for thrombolytic therapy. However, the need for emergency intervention such as surgical clipping or decompressive surgery would be less likely unless there is evidence of hemorrhagic transformation or impending herniation. Given the patient's vital signs, particularly the elevated blood pressure, intravenous labetalol therapy could be considered to manage hypertension, which is a common complication in ischemic stroke patients. The patient does not appear to have a contraindication for thrombolytic therapy based on the CT findings alone. Therefore, the most appropriate next step would typically be intravenous alteplase therapy if the patient meets the criteria for thrombolysis. </think>
<query> intravenous alteplase stroke management guidelines </query>
<retrieve> (1) Alteplase is a thrombolytic agent that is manufactured by recombinant DNA technology. **It is FDA approved for use in acute ischemic stroke,** pulmonary embolism, acute myocardial infarction, and occluded catheters. Off-label indications include catheter-directed thrombolysis in the treatment of peripheral arterial occlusive disease and deep vein thrombosis. This activity outlines the indications, contraindications, activity, adverse events, and toxicity of alteplase in the clinical setting as relates to the essential points necessary for members of an interprofessional team managing the care of patients in need of thrombolytic therapy. (2) Alteplase is a fibrinolytic agent; it also is referred to as tissue plasminogen activator (tPA). Alteplase converts plasminogen to the proteolytic enzyme plasmin, which lyses fibrin as well as fibrinogen. (3) Intravenous alteplase is cleared primarily by the liver with an initial half-life of fewer than 5 minutes and a terminal half-life of 72 minutes. When alteplase 2 mg is instilled into occluded catheters to restore catheter function, it is unlikely that plasma will attain pharmacologic concentrations of alteplase. </retrieve>
<think> The most appropriate next step in this patient's management would be Intravenous alteplase therapy. </think>
<answer> D. Intravenous alteplase therapy </answer>

Figure 12: Example of MED-RWR's Rollout Process: Confirmation-Driven Retrieval.

images with high similarity. In this example, a similar case depicting a meniscal tear is retrieved, accompanied by a descriptive caption. By integrating this highly pertinent multimodal information, the model was able to rectify its initial reasoning and correctly identify the correct answer. This case demonstrates that CDIR could overcome situations where the initial knowledge base is insufficient, thereby enhancing diagnostic accuracy.

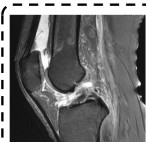 A 30-year-old tennis player presents with an injury shown in Figure A and is evaluating nonoperative management options for the affected structure. Among the following clinical findings, which one has the strongest correlation with the future development of arthritic changes when treated nonoperatively?
A. Positive Ober test
B. Tenderness over MCL origin without opening on valgus
C. Presence of pivot shift
D. Grade IIB Lachman
**E. Painful pop on McMurray test**

<think>The MRI shows a tear involving the anterolateral corner of the knee, which typically includes the lateral meniscus, anterior cruciate ligament (ACL), and/or the posterolateral ligaments. The presence of a significant tear in this area is associated with higher rates of osteoarthritis development compared to isolated tears of the ACL or medial collateral ligament. The Ober test is used to assess hip abduction, not knee stability or meniscal pathology. The Lachman test is specific for assessing ACL integrity, but its grade does not directly correlate with the future development of arthritic changes. The McMurray test is used to assess meniscal tears, and while positive, it suggests a meniscal tear rather than a meniscal injury. Tenderness over the MCL origin without opening on valgus stress is more indicative of a medial collateral ligament strain or sprain, which generally has better outcomes and lower risk of developing arthritic changes compared to a lateral meniscus tear. </think>

<query> clinical presentation knee meniscal tear arthritis progression </query>

**Less Informative Retrieval**

<retrieve> **Gonococcal arthritis** accounts for up to 70% of episodes of infectious arthritis. True gonococcal septic arthritis is less common than disseminated gonococcal infection (DGI) and always follows disseminated gonococcal arthritis. Disseminated gonococcal arthritis is unrecognized in one-third of patients. A single joint such as the hip, knee, ankle, or wrist is usually affected. Synovial fluid in affected joints contains more than 50,000 leukocytes per liter. Initial treatment involves ceftriaxone at a dosage of 1 g administered intravenously or intramuscularly every 24 hours. </retrieve>

<think> The Ober test is not specifically related to the lateral meniscus tear but is more commonly used for hip flexion/extension testing. The grade II Lachman test indicates a partial tear of the ACL, which is less likely to lead to osteoarthritis than a significant meniscal tear. The McMurray test is used to assess for meniscal tears, which would not be as closely correlated with the development of arthritis as a lateral meniscus tear. The strongest correlation with the future development of arthritic changes when treated nonoperatively is most likely with **the presence of a significant ACL tear**, which can be inferred from the MRI findings. This tear often leads to instability and degenerative changes over time unless surgically addressed. </think>

**Inaccurate Prediction and Low Confidence**

<answer> D. Grade IIB Lachman </answer> ✗     Confidence: 0.66 🔻

Figure 13: Example of Model Rollout Process without CDIR.

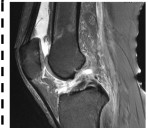 A 30-year-old tennis player presents with an injury shown in Figure A and is evaluating nonoperative management options for the affected structure. Among the following clinical findings, which one has the strongest correlation with the future development of arthritic changes when treated nonoperatively?
A. Positive Ober test
B. Tenderness over MCL origin without opening on valgus
C. Presence of pivot shift
D. Grade IIB Lachman
**E. Painful pop on McMurray test**

<think>The MRI shows a tear involving the anterolateral corner of the knee, which typically includes the lateral meniscus, anterior cruciate ligament (ACL), and/or the posterolateral ligaments. The presence of a significant tear in this area is associated with higher rates of osteoarthritis development compared to isolated tears of the ACL or medial collateral ligament. The Ober test is used to assess hip abduction, not knee stability or meniscal pathology. The Lachman test is specific for assessing ACL integrity, but its grade does not directly correlate with the future development of arthritic changes. The McMurray test is used to assess meniscal tears, and while positive, it suggests a meniscal tear rather than a meniscal injury. Tenderness over the MCL origin without opening on valgus stress is more indicative of a medial collateral ligament strain or sprain, which generally has better outcomes and lower risk of developing arthritic changes compared to a lateral meniscus tear. </think>

<query> clinical presentation knee meniscal tear arthritis progression </query>

<retrieve> **Gonococcal arthritis** accounts for up to 70% of episodes of infectious arthritis. True gonococcal septic arthritis is less common than disseminated gonococcal infection (DGI) and always follows disseminated gonococcal arthritis. Disseminated gonococcal arthritis is unrecognized in one-third of patients. A single joint such as the hip, knee, ankle, or wrist is usually affected. Synovial fluid in affected joints contains more than 50,000 leukocytes per liter. Initial treatment involves ceftriaxone at a dosage of 1 g administered intravenously or intramuscularly every 24 hours. </retrieve>

<think> The Ober test is not specifically related to the lateral meniscus tear but is more commonly used for hip flexion/extension testing. The grade II Lachman test indicates a partial tear of the ACL, which is less likely to lead to osteoarthritis than a significant meniscal tear. The McMurray test is used to assess for meniscal tears, which would not be as closely correlated with the development of arthritis as a lateral meniscus tear. The strongest correlation with the future development of arthritic changes when treated nonoperatively is most likely with the presence of a significant ACL tear, which can be inferred from the MRI findings. This tear often leads to instability and degenerative changes over time unless surgically addressed. </think>

**Image Re-retrieval**

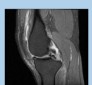 <retrieve> This MR image, presented in a sagittal view of the left knee, depicts the internal structures with good contrast. The focal point of the image reveals notable changes in the posterior horn of the medial meniscus, manifesting as an abnormal signal, likely indicating a **meniscal tear**. Above the tibia, part of the meniscus appears displaced superiorly, lying just beneath the posterior cruciate ligament (PCL), forming an imperfect alignment with typical knee anatomy. This altered position of the meniscal fragment suggests a meniscal flip or dislocation, which is abnormal. </retrieve>

<think> The MRI image shows a **meniscal tear**, which aligns with clinical findings such as a painful pop on **McMurray test** or a positive pivot shift. Among the given options, the presence of a painful pop on McMurray test (option E) is closely associated with the risk of developing degenerative joint disease due to chronic instability, which may lead to future arthritic changes. The McMurray test's strong correlation with degenerative joint disease underscores its importance in predicting long-term outcomes. Therefore, option E is the most likely correlate of future arthritic changes when treated nonoperatively. </think>
<answer> E. Painful pop on McMurray test </answer> ✓

Figure 14: Example of Model Rollout Process with CDIR.

## F.3 ABLATIVE CASE STUDIES

We provide ablative case studies in Figure 15 to demonstrate the effectiveness of the query semantic reward. While applying only the query semantic reward for textual alignment successfully helps the model identify the correct answer, the generated reasoning reveals an uninterpretable logical process. The generated query, "abdominal pain pneumoperitoneum" is a direct combination of the patient's symptom from the question and one of the potential answers, failing to explain why the specific diagnosis was investigated. In contrast, the model guided by the query semantic reward for multimodal alignment demonstrates a more effective and interpretable reasoning process. The multimodal alignment incentivizes queries that not only originate from a key visual observation but also relate it to textually relevant knowledge.

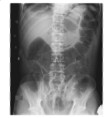 Examine the radiographic findings of a 56-year-old male presenting with abdominal pain, what is the most likely diagnosis?
A. Colonic carcinoma
B. Acute pancreatitis
C. Mesenteric ischemia
**D. Pneumoperitoneum**
E. Colonic volvulus

**w/o Query Semantic Reward**

\<think\> To determine the most likely diagnosis based on the radiographic findings, let's analyze the image: The image shows a lateral abdominal X-ray. There is a clear presence of air in the bowel loops, which is not typically seen in solid organ diseases like colonic carcinoma or acute pancreatitis. Given the presence of gas in the bowel loops and absence of other specific signs, pneumoperitoneum seems less likely than volvulus. \</think\>

\<query\> **colonic volvulus lateral abdominal X-ray appearance** \</query\>

\<retrieve\> (1) Volvulus occurs when a segment of the intestine twists around itself and its supporting mesentery, causing bowel obstruction and potentially cutting off the blood supply, which can lead to bowel ischemia. The condition presents with symptoms such as abdominal distension, pain, vomiting, constipation, and possibly bloody stools. If untreated, volvulus can progress to bowel necrosis. (2) Colonic volvulus, particularly affecting the sigmoid and cecum, is a potentially life-threatening condition requiring prompt diagnosis and intervention. While some cases, like sigmoid volvulus, may be treated with non-surgical methods like decompression with a rectal tube, cecal volvulus often necessitates surgical intervention. (3) The present topic covers volvulus in adults with specific differences from midgut volvulus in children. However, a detailed discussion of malrotation and midgut volvulus is beyond the scope of this topic. \</retrieve\>

\<think\> The image shows gas in the bowel loops, but no clear evidence of free air under the diaphragm, which would be more suggestive of pneumoperitoneum. \</think\>

\<answer\> E. Colonic volvulus \</answer\>

**w/ Text-level Query Semantic Reward**

\<think\> The patient's clinical presentation of abdominal pain and the radiographic findings, which show air in the retroperitoneal space. This condition typically presents with sudden onset of severe abdominal pain and is often associated with trauma or perforation of hollow organs. \</think\>

\<query\> **abdominal pain pneumoperitoneum** \</query\>

\<retrieve\> (1) Pneumoperitoneum is the space located behind the peritoneal cavity of the abdomen. The presence of air or gas in the abdominal cavity is often due to perforation of a hollow organ. (2) Peritoneum indicates the membrane lining the abdominal cavity and covering the abdominal organs. This thin, transparent serous membrane consists of two layers: the parietal peritoneum, which lines the inner walls of the abdomen, and the visceral peritoneum, which wraps around individual organs such as the stomach, liver, and intestines. (3) Free air under the diaphragm may not be visible on an erect chest x-ray, but CT will show pneumoperitoneum and fluid around the anastomosis. In patients who have undergone anterior resection, it is advised to perform the scan with rectal contrast. \</retrieve\>

\<think\> The patient's abdominal pain and the presence of air in the retroperitoneal space are not typical for colonic carcinoma or acute pancreatitis, as these conditions usually do not present with such specific radiographic findings. Mesenteric ischemia would typically show changes in bowel wall thickness rather than free air in the retroperitoneum. Colonic volvulus can cause similar symptoms but usually shows more localized changes on imaging. Pneumoperitoneum, characterized by free air in the peritoneal cavity, is a common finding in cases of perforation or trauma and aligns well with the clinical presentation and radiographic findings observed here. \</think\>

\<answer\> D. Pneumoperitoneum \</answer\>

**w/ Image & Text-level Query Semantic Reward**

\<think\> The radiograph shows free air under the diaphragm. This is typically associated with perforation of hollow organs such as the colon or stomach. Given the clinical context of abdominal pain, this finding is most consistent with a colonic perforation, which could be due to ischemia, carcinomatous stricture, or volvulus leading to necrosis. The absence of a visible mass rules out mesenteric ischemia as a primary cause, but it could be a secondary complication. \</think\>

\<query\> **radiographic signs colonic perforation** \</query\>

\<retrieve\> (1) **Perforation of a hollow viscus often causes pneumoperitoneum**, allowing air to escape from the bowel and accumulate below the diaphragm. The most common cause of pneumoperitoneum is perforation of a hollow viscus. Chest X-ray is the best projection to demonstrate pneumoperitoneum. (2) Perforation can lead to acute abdomen, septic shock, peritonitis, and other life-threatening conditions.. The presence of pneumatosis intestinalis with portal venous gas usually correlates with bowel ischemia in approximately 70% of cases. Extraintestinal complications are adhesions and compression of nearby structures. Cysts can cause outpouchings that directly compress adjacent organs. (3) Perforation of the stomach is a full-thickness injury of the wall of the organ. Since the peritoneum completely covers the stomach, perforation of the wall creates a communication between the gastric lumen and the peritoneal cavity. If the perforation occurs acutely, there is no time for an inflammatory reaction to wall off the perforation, and the gastric content is free to enter the general peritoneal cavity, causing chemical peritonitis. Perforations occurring over a prolonged period may be contained locally by the inflammatory reaction. \</retrieve\>

\<think\> The most likely diagnosis based on the radiographic findings is D. Pneumoperitoneum, which often leads to the clinical presentation described. Further clinical correlation is necessary for definitive diagnosis and appropriate management. \</think\>

\<answer\> D. Pneumoperitoneum \</answer\>

Figure 15: Ablative Case Studies of the Query Semantic Reward.

## F.4 FAILURE CASE ANALYSIS

To gain a deeper understanding of the behavior and capabilities of our proposed MED-RWR, we conduct additional analysis of failure cases to identify potential limitations and illuminate future

directions for enhancement. We generalize the failures into three categories: (1) *Retrieval Failure* (Figure 16): the retriever returns irrelevant information despite an effective query; (2) *Integration Failure* (Figure 17): the model fails to incorporate the helpful retrieved knowledge into the reasoning process, adhering to the original reasoning path. (3) *Perceptual Failure* (Figure 18): the model's insufficient internal knowledge about specific conditions leads to the misinterpretation of the medical scan. These cases indicate that our proposed framework can extend the model's upper bound by encouraging the model to actively retrieve external knowledge. However, enriching the base MLLM with more extensive medical knowledge while enhancing the retriever's ability to identify relevant information from extensive knowledge is also crucial to unlock the full potential of reasoning-with-retrieval capability.

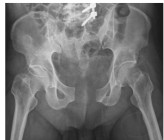

A 45-year-old man presented to the emergency department after a motorcycle accident. Physical examination was notable for a bruise in the pubic area and severe pain in the anterior and posterior pelvic areas with manual compression. What is the diagnosis?
A. Avascular necrosis of the hip
B. Hip dysplasia
C. Femoral head fracture
D. Open-book fracture
E. Straddle pelvic fracture

\<think\> The radiograph shows **a disruption of the pelvis with displacement of the pubic rami, which is characteristic of an open-book fracture**. This type of fracture typically occurs when there is a force applied to the pelvis, causing the pubic rami to spread apart. The presence of a fracture line across the symphysis pubis and the displacement of the pubic rami supports this diagnosis. There is no evidence of avascular necrosis, femoral head fracture, or hip dysplasia. The physical examination findings of severe pain with manual compression also support the diagnosis of an open-book fracture. \</think\>
\<query\> radiographic features **open-book fracture** \</query\>
\<retrieve\> (1) An **open fracture**, also known as a compound fracture, is a high-energy injury where a broken bone is exposed through the skin. This breach of the skin and underlying soft tissue creates a direct pathway for bacteria from the environment to contaminate the fracture site, leading to a significantly increased risk of infection, including cellulitis and osteomyelitis (bone infection). The management of open fractures is a surgical emergency. Immediate treatment involves the administration of intravenous antibiotics and a tetanus booster if needed. (2) **Fractures of the mandibular condyle** are common facial injuries, often resulting from falls, assaults, or motor vehicle accidents. On radiographic evaluation, non-displaced fractures can sometimes be subtle and are best viewed on an anteroposterior (AP) or Towne's view projection. However, for displaced fractures, the characteristic deformity is dictated by muscle pull. The lateral pterygoid muscle, which attaches to the neck of the condyle, contracts and pulls the fractured condylar head anteriorly and medially. This action results in a classic radiographic sign where the apparent width of the fractured condyle seems increased on imaging. (3) Clinically, patients may present with pain in front of the ear, swelling, malocclusion (teeth not fitting together properly), and deviation of the jaw toward the side of the fracture upon opening. While plain radiographs can be used for initial assessment, a CT scan or a panoramic radiograph (Panorex) is often preferred for a more detailed evaluation of the fracture pattern and degree of displacement.
\</retrieve\>
\<think\> Based on the radiographic findings and clinical presentation, the most likely diagnosis is D. Open-book fracture.\</think\>
\<answer\> D. Open-book fracture \</answer\>

Figure 16: Example of Failure Case: Retrieval Failure

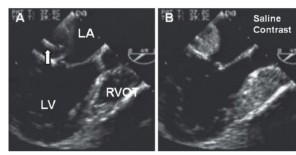

This patient has:
A. Dilated coronary sinus and dextrocardia
B. Dilated coronary sinus and levocardia
C. Cor triatriatum
D. Aneurysm of circumflex coronary artery

\<think\> The images show a dilated coronary sinus with saline contrast filling the right atrium and ventricle, which is consistent with a **dilated coronary sinus**. The presence of saline contrast in the right side of the heart suggests a communication between the coronary sinus and the right atrium or ventricle. \</think\>
\<query\> Dilated coronary sinus echocardiography findings clinical significance \</query\>
\<retrieve\> (1) Dilated coronary sinus can be indicated by opacification at the AV groove with saline contrast. Anomalous connection of the superior vena cava into the coronary sinus can lead to opacification with saline contrast. Presence of a pacemaker lead in the coronary sinus can be noted in imaging. Dextrocardia is indicated when the left ventricular outflow tract is on the right side during imaging. **A 49 degree angle imaging suggests dextrocardia when the left ventricular outflow tract is observed on the right side. Imaging at 120–140 degrees is typical for levocardia, presenting a sort of mirror image.** (2) A dilated coronary sinus can indicate an anomalous connection of the superior vena cava into the coronary sinus. Dextrocardia is characterized by the left ventricular outflow tract being positioned on the right side. Saline contrast opacification at the atrioventricular groove can help identify anomalous venous connections. In cardiac imaging, certain angles can help distinguish between dextrocardia and levocardia. (3) The coronary sinus can become dilated due to increased flow or pressure. Persistent left superior vena cava can result in a dilated coronary sinus. Congestive heart failure can result in a dilated coronary sinus. Unroofed coronary sinus can result in a dilated coronary sinus. The coronary sinus is located in the atrioventricular groove. The descending aorta is located in the posterior mediastinum and is extrapericardial. \</retrieve\>
\<think\> The patient's imaging shows a dilated coronary sinus with saline contrast filling the right atrium and ventricle, which is consistent with a dilated coronary sinus. **There is no evidence of dextrocardia or levocardia**, nor does the image suggest a cor triatriatum or an aneurysm of the circumflex coronary artery. Therefore, the most likely diagnosis is B. Dilated coronary sinus and levocardia.\</think\>
\<answer\> B. Dilated coronary sinus and levocardia \</answer\>

Figure 17: Example of Failure Case: Integration Failure

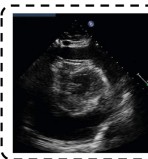 A 45-year-old female was admitted to the hospital with complaints of acute onset of shortness of breath. This still frame shows:
**A. Pleural effusion**
B. Pericardial effusion
C. Normal heart

<think> This ultrasound image shows **a large, homogenous, hypoechoic area surrounding the cardiac silhouette**, which is consistent with pericardial effusion. The presence of a pericardial effusion can lead to symptoms such as shortness of breath due to compression of the heart and reduced cardiac output. </think>
<query> echocardiography pericardial effusion clinical presentation </query>
<retrieve> (1) Pericardial effusion can be identified using transesophageal echocardiogram (TEE). Chest pain may require further imaging to determine the underlying cause.\n(2) A large pericardial effusion can compress the right heart. Cardiac tamponade physiology involves significant hemodynamic changes often due to pericardial effusion. Subcostal view in echocardiography can be used to identify pericardial effusion and its effects on the heart.\n(3) Large pericardial effusions have a depth greater than 2 cm. In echocardiography, large pericardial effusions are seen primarily in the near field of the parasternal long-axis view. Pericardial effusions greater than 2 cm in depth are categorized as large. Echocardiographic signs of tamponade physiology include serpentine movement of the right ventricular free wall and diastolic right ventricular collapse. Right ventricular hypertrophy can be due to chronically elevated pulmonary artery pressure. Right ventricular collapse may occur in mid/late diastole in the presence of right ventricular hypertrophy instead of early diastole. </retrieve>
<think> Based on the retrieved information, **the appearance in this image aligns with large pericardial effusion**, which is consistent with option B. </think>
<answer> B. Pericardial effusion </answer>

Figure 18: Example of Failure Case: Perceptual Failure

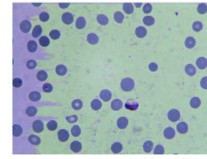 A 6-year-old girl is brought to the clinic due to fatigue, pallor, and intermittent fever. She recently returned from a trip abroad. Upon evaluating her blood smear, there are numerous inclusions within the erythrocytes. Based on the scatter diagram and the blood smears, what potential diagnosis is indicated by the findings in the highlighted region of the scatter diagram?
**A. Malaria due to Plasmodium falciparum**
B. Sickle cell anemia
C. Bacterial infection such as Streptococcus pneumoniae
D. Viral infection causing cytomegalovirus

<think>The image provided shows red blood cells (RBCs) with visible cytoplasmic inclusions characteristic of parasitic infestation. These inclusions are indicative of red blood cell parasites typically seen in malaria species. Plasmodium falciparum, a common cause of malaria in regions where the disease is imported, is known for invading RBCs and producing these distinctive inclusions, often seen in clinical blood smears. </think>
<query> Malaria parasite infections blood cells inclusions malaria </query>
<retrieve> (1) malaria infection\tMalaria infection is a disease caused by Plasmodium parasites, transmitted to humans through the bite of infected Anopheles mosquitoes. Key characteristics visible in medical images include red blood cells infected with Plasmodium, often appearing enlarged or showing inclusions such as ring-shaped parasites in blood smears. Symptoms include fever, chills, and flu-like illness, and without treatment, severe complications like organ failure can occur.
(2) malaria\tA disease caused by Plasmodium parasites, transmitted by the bite of infected mosquitoes.
(3) Malaria pathophysiology and its effects on red blood cells and the body.\tIn malaria caused by Plasmodium vivax, the size of red blood cells (RBCs) is increased. </retrieve>
<think>The observed cytoplasmic inclusions within erythrocytes are characteristic of Plasmodium spp., with Plasmodium falciparum being particularly notable for causing these types of infections in regions where malaria is endemic or imported.</think>
<query> Plasmodium falciparum blood red blood cells inclusion </query>
<retrieve> (1) falciparum\tA species of Plasmodium that causes the most severe form of malaria.
(2) P. falciparum\tThe most dangerous of the malaria parasites that infect humans, known for causing severe and potentially fatal illness.
(3) Plasmodium falciparum\tA species of the Plasmodium parasite that causes the most severe form of malaria in humans. </retrieve>
<think> The findings are consistent with malaria. Other options such as bacterial infections or cytomegalovirus typically do not cause these RBC structures. Sickle cell anemia involves cell deformation but no cytoplasmic inclusions like those shown.</think>
<answer>**A. Malaria due to Plasmodium falciparum**</answer>

Figure 19: Example of Multi-Hop Retrieval

## F.5 DISCUSSION ON MULTI-HOP RETRIEVAL

Our model demonstrates multi-hop retrieval capacity, as illustrated in Figure 19. In these multi-turn scenarios, the model generates queries across multiple reasoning steps, with each subsequent turn conditioned on the full context of previous reasoning steps and retrieved documents. While our method shows preliminary capability in certain multi-turn cases, developing robust multi-hop reasoning is a promising direction for extending this proactive retrieval pipeline to more complicated scenarios requiring iterative clinical reasoning and evidence accumulation, such as rare disease diagnosis and differential diagnosis. In future work, this problem can potentially be solved by incorporating medical knowledge graphs to construct more complex training data dedicated for multi-hop retrieval and reflection during thinking.

