# OpenReview forum: "Proactive Reasoning-with-Retrieval Framework for Medical Multimodal Large Language Models"
_ICLR.cc/2026/Conference — Submitted to ICLR 2026_

### Official Review · Reviewer_Hige · 2025-10-21

**Soundness:** 3
**Presentation:** 3
**Contribution:** 2
**Rating:** 6
**Confidence:** 4

**Summary:**

This paper proposes Med-RWR, a multimodal Reasoning-with-Retrieval framework for the medical VQA task. Med-RWR first develops an environment with a curated multimodal dataset and medical knowledge base to train the medical MLLM to use external reliable sources. Then, it introduces a two-stage (text-only to multimodal) RL training strategy to guide the model to retrieve visually and textually related references (semantic reward) and gain confidence through retrieving (confidence gain reward). During inference, it introduces a Confidence-Driven Image Re-retrieval strategy if low confidence is detected.

**Strengths:**

1. The idea of reasoning-with-retrieving is sound, with support from reliable medical KB construction and reward designs (semantic reward and confidence gain reward).
2. The paper is overall well-written, with each component clearly explained with visualizations.
3. The experimental results show that Med-RWR effectively enhances performance across different medical VQA benchmarks.

**Weaknesses:**

1. Misconception on Test-Time Scaling (TTS). This paper mentions that the Confidence-Driven Image Retrieval (CDIR) method is for test-time scaling. However, this might be a misconception. Will retrieving more similar cases further improve performance? Providing a figure on this would be helpful. If the answer is yes, then it is fine to use TTS. Otherwise, it would be better to use another terminology for clarity.

2. Too many hand-crafted hyperparameters. For example, the reward weights in Eq. 4 and the confidence threshold in line 471.

3. Lack of baselines. This work incorporates multimodal RAG with reasoning to enhance medical MLLMs' capability. However, it is mainly compared to reasoning medical MLLM. How does it perform compared to multimodal RAG methods [1, 2]?


[1] MMed-RAG: Versatile Multimodal RAG System for Medical Vision Language Models. ICLR 2025.

[2] RULE: Reliable Multimodal RAG for Factuality in Medical Vision Language Models. EMNLP 2024.

**Questions:**

Please see weaknesses. I would be willing to adjust my ratings if the concerns are addressed.

---

> ### Author Response · Authors · 2025-11-25
> **Response to Reviewer Hige [Part I]**
>
> We are grateful for your recognition of our paper's soundness and novelty. Thanks for your efforts in the review and precious comments.
>
> ### **W1: "Misconception on Test-Time Scaling (TTS)..."**
>
> Thank you for your insightful comments in Test-Time Scaling (TTS). We respectfully believe that CDIR matches the definition of TTS. Test-Time Scaling is defined as methods that “increase the compute at test time to get better results. [1]” At test time, our model increase the compute by introducing image re-retrieval strategy to enrich the model input with more diverse information from similar medical cases. This computation increase got better performance (see Table 1 in the original paper).
>
> Following your suggestion, we further conducted a set of experiments to illustrate the scalability of CDIR mechanism.
>
> **The size of retrieved cases.** We have conducted ablation studies increasing the number of retrieved cases from 0 to 5. As shown in the tables below, the performance initially increases then decreases as the number of retrieved cases grows, indicating that retrieving more cases not necessarily provides consistent performance improvement. We attribute this to the diminishing similarity of increased samples. Retrieving more samples could expose the model to less similar information, which would potentially confuse the model with noisy information.
>
> | **CDIR Retrieved Case** | **MMMU-H&M** | **MMMU-Pro-H&M** | **MedXpertQA-MM** |
> | --- | --- | --- | --- |
> | 0 (w/o CDIR) | 65.5 | 43.7 | 27.2 |
> | 1 | 66.2 | 44.1 | 27.5 |
> | 2 | 66.2 | 45.5 | 27.7 |
> | 3 | 67.6 | 45.1 | 27.4 |
> | 4 | 66.9 | 44.1 | 27.5 |
> | 5 | 66.9 | 44.4 | 27.4 |
>
> **The size of retrieval corpus.** We further conduct experiment expanding the retrieval pool from 10,000 to 50,000 images, where we observe a consistent performance improvement. This demonstrates that the effectiveness of CDIR mechanism scales up with corpus size. A larger corpus increases the likelihood of finding highly similar cases, thus providing more relevant information to the model.
>
> | **CDIR Corpus Scale** | **MMMU-H&M** | **MMMU-Pro-H&M** | **MedXpertQA-MM** |
> | --- | --- | --- | --- |
> | 0 (w/o CDIR) | 65.5 | 43.7 | 27.2 |
> | 10,000 | 66.2 | 44.1 | 27.5 |
> | 20,000 | 66.2 | 44.4 | 27.7 |
> | 30,000 | 67.6 | 45.1 | 27.8 |
> | 40,000 | 68.3 | 44.8 | 27.6 |
> | 50,000 | **68.9** | **45.5** | **27.8** |
>
> We have included the figures and discussions in Figure 9 and Section E.7.
>
> [1] Muennighoff, Niklas, et al. "s1: Simple test-time scaling." *Proceedings of the 2025 Conference on Empirical Methods in Natural Language Processing*. 2025.
>
> ### **W2: "Too many hand-crafted hyperparameters..."**
>
> Thanks for your insightful question. We would like to claim that we set the hyperparameters following these intentions: (1) prioritizing the dominant task objective, the accuracy reward and (2) balancing the scale of other supporting rewards to promote stable training and not affect the primary target. Figure 8 in the Appendix shows that our chosen weights lead to a stable training process, where all reward components are rescaled into a similar and reasonable range. Specially, we intentionally use a smaller weight for Confidence Gain Reward to prevent the model from overly optimizing on the confidence scores, but instead focusing on the retrieved information quality.
>
> The confidence threshold is empirically determined by analyzing the confidence distribution on a small validation subset. Statistically, predictions with confidence above 0.8 have a higher likelihood to associate with a correct answer. Therefore, we adopt 0.8 as the threshold to distinguish confident predictions from uncertain ones.

---

> > ### Author Response · Authors · 2025-11-25
> > **Response to Reviewer Hige [Part II]**
> >
> > ### **W3: "Lack of baselines..."**
> >
> > Thanks for your suggestions on including more baselines.  MMed-RAG [1] and RULE [2] are medical multimodal RAG methods that used **image-based retrieval** with **modality-specific vision-language retrievers.** To fairly compare with their methods, we train our model with the same finetuning data and evaluate on the same test benchmarks as MMed-RAG [1] and RULE [2]. The results are shown below (Accuracy as the metric). Results of MMed-RAG and RULE are referenced from their original paper. The results demonstrated that our proposed reasoning-with-retrieval paradigm still has better performance over strong medical RAG baselines under identical experimental setup. We have updated Section E.6 in the revision to incorporate these discussions and experiments.
> >
> > | **Model** | **IU-Xray** | **MIMIC-CXR** | **Quilt-1M** | **PMC-OA Pathology** |
> > | --- | --- | --- | --- | --- |
> > | RULE | 87.84 | **83.92** | 68.97 | 61.41 |
> > | MMed-RAG | 89.54 | 83.57 | 72.95 | 64.54 |
> > | Med-RwR | **92.81** | **83.92** | **75.74** | **67.35** |
> >
> > [1] Xia, Peng, et al. "Mmed-rag: Versatile multimodal rag system for medical vision language models." *The Thirteenth International Conference on Learning Representations*. 2025.
> >
> > [2] Xia, Peng, et al. "Rule: Reliable multimodal rag for factuality in medical vision language models.” *Proceedings of the 2024 Conference on Empirical Methods in Natural Language Processing.* 2024.

---

### Official Review · Reviewer_Wkgq · 2025-10-31

**Soundness:** 3
**Presentation:** 3
**Contribution:** 3
**Rating:** 4
**Confidence:** 5

**Summary:**

This paper introduces MED-RWR, a novel framework for medical multimodal large language models (MLLMs) designed to address the problem of factual inaccuracies and hallucinations by actively integrating external knowledge during reasoning. Unlike existing methods that rely solely on internal knowledge or unimodal retrieval, MED-RWR enables the MLLM to proactively query for information based on both visual and textual inputs. The core of the method is a two-stage RL strategy. The framework is evaluated on several public medical benchmarks and a newly curated ECBench, demonstrating significant improvements in accuracy and generalizability to scarce domains.

**Strengths:**

- The paper tackles a critical and high-stakes problem in medical AI: the unreliability and hallucination of MLLMs, which stems from their reliance on static internal knowledge. The proposed reasoning-with-retrieval approach is a well-motivated solution.
- The reward engineering is a key strength. The Query Semantic Reward is novel for jointly encouraging textual relevance and visual grounding. Furthermore, the Confidence Gain Reward is an nice way to optimize for the utility of the retrieved information by measuring its impact on the model's confidence in the correct answer.
- The paper includes a comprehensive set of ablation studies that validate the key design choices.

**Weaknesses:**

- Lack the Ethics statement and Reproducibility statement in the main text.
- The CDIR mechanism, while interesting, has questionable scalability. At inference time, it computes image similarity against a randomly selected subset of 10,000 images from the multimodal corpus. This selection seems arbitrary, and the paper does not address how this method would scale to a more realistic, much larger corpus (e.g., PubMedVision corpus).
- The framework's retrieval mechanism is effectively limited to a single turn, which is insufficient for complex, multi-hop medical reasoning. The paper's own prompt design in Table 10 is contradictory, stating both "Only one query is allowed" and "Multiple think-query-retrieve cycles may occur". In practice, all case studies provided (e.g., Figures 9-14) demonstrate only a single query-retrieval step. This single-turn design is brittle. If the initial retrieval is inaccurate (as shown in the failure case in Figure 15), the model has no mechanism to correct its course by re-querying. This lack of iterative reasoning may explain why the overall performance gains are not as substantial as one might expect from a more sophisticated multi-step agent.
- Lack a section of discussion of recent retrieval-based methods [1,2,3,4] and Reasoning-with-Retrieval methods [5,6,7].

[1] MMed-RAG: Versatile Multimodal RAG System for Medical Vision Language Models. ICLR 2025.

[2] Fact-Aware Multimodal Retrieval Augmentation for Accurate Medical Radiology Report Generation. NAACL 2025.

[3] RULE: Reliable Multimodal RAG for Factuality in Medical Vision Language Models. EMNLP 2024.

[4] Patho-agenticrag: Towards multimodal agentic retrieval-augmented generation for pathology vlms via reinforcement learning. arXiv preprint arXiv:2508.02258.

[5] MMSearch-R1: Incentivizing LMMs to Search. arXiv preprint arXiv:2506.20670.

[6] Webwatcher: Breaking new frontier of vision-language deep research agent. arXiv preprint arXiv:2508.05748.

[7] DeepMMSearch-R1: Empowering Multimodal LLMs in Multimodal Web Search. arXiv preprint arXiv:2510.12801.

**Questions:**

- See weaknesses.
- If the authors can address the limitations mentioned in the weaknesses, I will consider raising my score.

**Details Of Ethics Concerns:**

There is a requirement on Ethics statement this year. But this paper lacks this part. And it proposes a benchmark called ECBench. It should contain a section to discuss the ethics concerns.

---

> ### Author Response · Authors · 2025-11-25
> **Response to Reviewer Wkgq [Part I]**
>
> We are grateful for the reviewer's appreciation of our targeted problem and design choices. Thanks for your time in the review process and the insightful comments.
>
> ### **W1: "Lack the Ethics statement and Reproducibility..."**
>
> Thanks for your suggestion. We have included the required sections accordingly. Regarding the in-house dataset ECBench, it is collected from authoritative echocardiography practice books, which is used under an appropriate copyright license and does not involve any patient data or raise ethical concerns.
>
> ### **W2: "The CDIR mechanism, while interesting, has questionable scalability...**
>
> Thank you for your insights regarding the scalability of CDIR. We agree that applying CDIR to larger‑scale corpora is the ultimate goal, and we believe our approach can be straightforwardly scaled to datasets such as the PubMedVision corpus.
>
> **Why we use 10,000 images.** We selected 10,000 images to ensure computational feasibility for validating the core CDIR mechanism. The main contribution of this work is introducing the novel concept of CDIR and demonstrating its effectiveness. Specifically, enriching model inference with similar multimodal cases improves predictions when confidence is low. Scaling to a larger multimodal corpus, such as PubMedVision, can be achieved by pre‑computing multimodal feature vectors offline and indexing them with [FAISS](https://github.com/facebookresearch/faiss) to efficiently retrieve similar cases from millions of candidates. However, such large‑scale expansion primarily requires additional computational resources and engineering effort rather than methodological innovation. As this lies outside the scope of the present study, we leave it for future work.
>
> **Scalability is straightforward.** Our results show that CDIR remains effective even with a relatively small multimodal corpus of 10,000 images. Moreover, increasing the retrieval pool from 10,000 to 50,000 images yields consistent performance gains. This demonstrates that the effectiveness of CDIR is positively correlated with the size of the retrieval corpus. Thus, rather than being limited by scalability, CDIR is further strengthened as the size of the retrieval corpus grows. We have incorporated these discussions in the revised paper in Section E.7.
>
> | **CDIR Corpus Scale** | **MMMU H&M** | **MMMU-Pro H&M** | **MedXpertQA-MM** |
> | --- | --- | --- | --- |
> | 0 (w/o CDIR) | 65.5 | 43.7 | 27.2 |
> | 10,000 | 66.2 | 44.1 | 27.5 |
> | 20,000 | 66.2 | 44.4 | 27.7 |
> | 30,000 | 67.6 | 45.1 | 27.8 |
> | 40,000 | 68.3 | 44.8 | 27.6 |
> | 50,000 | **68.9** | **45.5** | **27.8** |
>
> ### **W3: "The framework's retrieval mechanism is effectively limited to a single turn, which is insufficient for complex, multi-hop medical reasoning...**
>
> Thanks for your constructive comment for retrieval mechanism.
>
> **Clarification on the Multi-Turn Design and Capacity.** Firstly, we would like to clarify that the prompt design in Table 10 supports an iterative process: **multi-turn reasoning** with **one query in each turn.** In each turn of this process, the model is permitted to generate a single search query. In multi-turn circumstances, the contexts from all previous turns, including the history of reasoning steps and retrieved documents, will be concatenated as the input for any subsequent turn. Thus, the whole process may contain multiple turns, formulating think-query-retrieve cycles. **In turn, the entire mechanism supports multi-turn reasoning and re-querying.** Following your suggestion, we added example for multi-turn process in Figure 19, demonstrating this multi-turn capacity.
>
> **Current Scope and Future Direction for Multi-Hop Reasoning.** We acknowledge your observation that the model predominantly uses single-turn retrieval in practice, with multi-turn retrieval occurring in 0.4-1.3% of cases across benchmarks. However, this aligns with our target on **general medical scenarios** where single-hop reasoning-with-retrieval paradigm *effectively boosts model performance*. Our results show substantial improvements across multiple medical VQA benchmarks (3.7% on MMMU-Pro-H&M and 8% on ECBench compared with SOTA), demonstrating our approach successfully grounds clinical reasoning in reliable external knowledge and reduces hallucinations through proactive retrieval, which is *currently underexplored in existing models*. We agree that the mentioned failure case suggests a promising direction to develop robust multi-hop reasoning, where the model can iteratively query from alternative perspectives when initial retrieval is ineffective. This capability has the potential to extend our proactive retrieval pipeline to more complex scenarios, such as rare diseases and differential diagnosis. In future work, this problem can potentially be solved by incorporating medical knowledge graphs to construct more complex training data dedicated for multi-hop retrieval. We have included these discussions in the revised paper in Section F.5.

---

> > ### Author Response · Authors · 2025-11-25
> > **Response to Reviewer Wkgq [Part II]**
> >
> > ### **W4: "Lack a section of discussion of recent retrieval-based methods and Reasoning-with-Retrieval methods."**
> >
> > Thanks for your suggestion. We have included the discussions in Section 2. The added discussion is provided below for your reference.
> >
> > **Discussion of  Retrieval-Based Methods.** To expose models to additional clinical evidence from external databases, Retrieval-Augmented Generation (RAG) has been applied to enhance medical models. RULE [3] addresses factual errors in medical MLLMs through calibrated retrieval context selection and preference-based fine-tuning, balancing model's reliance on inherent knowledge with external information. MMed-RAG [1] advances this approach by introducing a more versatile multimodal RAG system that employs domain-aware retrieval and adaptive context selection. [2] augments report generation with high-quality references with a fact-aware multimodal RAG pipeline. However, these medical RAG methods only support static retrieval, performing retrieval operations without the ability to adaptively retrieve additional information based on the model's generation process, which limits their effectiveness in handling complex medical problems. **Unlike these methods, we propose a novel  reasoning-with-retrieval paradigm that incentivizes the medical MLLM to proactively retrieve relevant knowledge during its reasoning process, allowing more efficient and targeted knowledge augmentation.**
> >
> > **Discussion of  Reasoning-with-Retrieval Methods.** To incentivize a more flexible paradigm, researchers have explored integrating retrieval into the reasoning process to enable mutual enhancement. In the multimodal domain, [5] and [7] equip MLLMs with agentic capabilities of adaptive retrieval from Internet sources. WebWatcher [6] enhances MLLMs' visual-language reasoning capabilities by incorporating various tools such as web search and  optical character recognition (OCR). Despite these advances, there remains a critical gap in exploring multimodal reasoning with adaptive retrieval based on both imaging and text in the medical domain, where the retrieved knowledge often supports differential diagnosis reasoning among multiple hypotheses, rather than seeking a single correct answer. This requires the model to be aware of anatomical structures and pathological patterns in the medical scans, a capability that general text-based agentic search models lack. Although [4] combines a retrieval agent with a pathology MLLM to enable agentic multimodal retrieval from pathology textbooks, its decoupled architecture and restriction to pathology-specific knowledge limit its ability to perform integrated retrieval during reasoning across diverse medical modalities. **In contrast, we develop a two-stage reinforcement learning (RL) strategy with medically-tailored rewards to train the MLLM in an end-to-end manner with various modalities. Critically, our designed rewards enable the model to leverage both visual clues from medical scans and clinical information from textual contexts when formulating queries, ensuring more relevant and accurate information retrieval.**
> >
> > [1] Xia, Peng, et al. "Mmed-rag: Versatile multimodal rag system for medical vision language models." *The Thirteenth International Conference on Learning Representations*. 2025.
> >
> > [2] Sun, Liwen, et al. "Fact-aware multimodal retrieval augmentation for accurate medical radiology report generation." *Proceedings of the 2025 Conference of the Nations of the Americas Chapter of the Association for Computational Linguistics: Human Language Technologies*. 2025.
> >
> > [3] Xia, Peng, et al. "Rule: Reliable multimodal rag for factuality in medical vision language models.” *Proceedings of the 2024 Conference on Empirical Methods in Natural Language Processing.* 2024.
> >
> > [4] Zhang, Wenchuan, et al. "Patho-agenticrag: Towards multimodal agentic retrieval-augmented generation for pathology vlms via reinforcement learning." *arXiv preprint arXiv:2508.02258* (2025).
> >
> > [5] Wu, Jinming, et al. "MMSearch-R1: Incentivizing LMMs to Search." *arXiv preprint arXiv:2506.20670* (2025).
> >
> > [6] Geng, Xinyu, et al. "Webwatcher: Breaking new frontier of vision-language deep research agent." *arXiv preprint arXiv:2508.05748* (2025).
> >
> > [7] Narayan, Kartik, et al. "DeepMMSearch-R1: Empowering Multimodal LLMs in Multimodal Web Search." *arXiv preprint arXiv:2510.12801* (2025).

---

### Official Review · Reviewer_fZCw · 2025-11-01

**Soundness:** 3
**Presentation:** 2
**Contribution:** 3
**Rating:** 6
**Confidence:** 3

**Summary:**

This paper proposes MED-RWR, a comprehensive Multimodal Medical Reasoning-with-Retrieval framework, to address the factual inaccuracies in existing medical MLLMs by encouraging MLLMs to proactively retrieve external knowledge using both visual and textual information during reasoning. The framework utilizes a two-stage reinforcement learning strategy with customized rewards to facilitate effective multimodal retrieval, and further introduces a Confidence-Driven Image Reretrieval (CDIR) mechanism to augment potentially insufficient information when the model's confidence is low.

**Strengths:**

1. The paper is well-written, with clear logic and is easy to understand.
2. Extensive experiments on multiple benchmarks demonstrate the superior performance of the proposed MED-RWR.
3. Comprehensive ablation analysis shows the effectiveness of each component.

**Weaknesses:**

1. Why does Equation 2 only compute the semantic similarity between the image and the query, instead of also considering the retrieved content as in Equation 1?
2. Section 3.1 mentions that the difficulty levels of the samples were stratified during dataset construction for progressive curriculum training, but curriculum training does not appear to be utilized subsequently in the methodology.
3. In line 309, it is mentioned that “We apply accuracy and format rewards to instill the model’s fundamental medical reasoning-with-retrieval capabilities.” How is the retrieval capability obtained in this context?
4. Based on my understanding, the last row in Table 2 should represent the complete MED-RWR model. If so, why do the numerical values not correspond to those reported in Table 1?
5. The base model that MED-RWR was fine-tuned upon is not specified.
6. What knowledge base did the Agentic Search MLLM (used in the comparison methods) utilize for external knowledge retrieval?
7. Some presentations need optimization. For example, Figure 1 is somewhat cluttered and a bit hard to interpret, while the headers in Tables 2 and 3 appear to be misaligned.

**Questions:**

Please refer to the Weaknesses.

---

> ### Author Response · Authors · 2025-11-25
> **Response to Reviewer fZCw**
>
> We are deeply grateful for your recognition of our paper's contribution and performance. Thanks for your constructive comments.
>
> **W1: "Why does Equation 2 only compute the semantic similarity between the image and the query, instead of also considering the retrieved content as in Equation 1?"**
>
> Thanks for your valuable question. The reason why we did not introduce retrieved contents to Equation 2 are two folded:
>
> **CLIP’s limitation in processing long documents**: CLIP-based model faces limitations to align an image with long documents [1]. The retrieved content often consists of lengthy paragraphs. As a result, the similarity score between the image and all retrieved contents are not always faithful, introducing unstable reward signals.
>
> **Direct image-query alignment**: Our primary goal for this reward is to encourage the model to generate **visually-grounded queries**. The retrieval process flows from image to query, and then from query to retrieved contents. Aligning the image with retrieved content is indirect since it would be confounded by the performance of the intermediate retriever module. Therefore, we directly calculate the similarity between the image and the query as the reward.
>
> [1] Urbanek, Jack, et al. "A picture is worth more than 77 text tokens: Evaluating clip-style models on dense captions." *Proceedings of the IEEE/CVF Conference on Computer Vision and Pattern Recognition*. 2024.
>
> **W2: "Section 3.1 mentions that the difficulty levels of the samples were stratified during dataset construction for progressive curriculum training, but curriculum training does not appear to be utilized subsequently in the methodology."**
>
> Thanks for your question. Our methodology does incorporate the stratified dataset for curriculum learning. We manually arrange the training dataset from easier to harder based on the difficulty labels. During training, the data loader processes the dataset in this sequential order, enabling the model to learn from simpler examples and then progress to more complex ones. We have clarified this implementation detail in Section 3.1 of the revised version.
>
> **W3: "In line 309, it is mentioned that “We apply accuracy and format rewards to instill the model’s fundamental medical reasoning-with-retrieval capabilities.” How is the retrieval capability obtained in this context?"**
>
> Thanks for your question. In the first stage of training, the retrieval behavior is guided with prompting and incentivized with format and accuracy reward. Specifically, the instruction shown in Table 10 in the Appendix encourages the model to retrieve when required, and the format reward will provide a reward signal when retrieval is activated and leads to a correct answer. Additionally, the accuracy reward inherently drives the model to retrieve meaningful contents that could benefit decision making.
>
> **W4: "Based on my understanding, the last row in Table 2 should represent the complete MED-RWR model. If so, why do the numerical values not correspond to those reported in Table 1?"**
>
> Thanks for your question. We would like to clarify that the last row in Table 2 does not represent the complete Med-RwR model, since it does not include Query Semantic Reward and Confidence Gain Reward. The purpose of Table 2 is to assess the impact of the training stages. To ensure a fair comparison, we maintained the reward function constant across all the experiments in the table, where only Accuracy and Format rewards are applied. Therefore, the model in the last row of Table 2 was trained for two stages with these two rewards, as illustrated in the table caption.
>
> **W5: "The base model that MED-RWR was fine-tuned upon is not specified."**
>
> Thanks for your suggestion. Med-RwR was finetuned from Qwen2.5-VL-7B. We have noted this detail in Section 4.1 of the revised version.
>
> **W6: "What knowledge base did the Agentic Search MLLM (used in the comparison methods) utilize for external knowledge retrieval?"**
>
> Thanks for your question. To ensure a fair comparison, we use the same knowledge base and retriever as our proposed framework for Agentic Search MLLM reproduction. We have claimed the implementation detail in Section D.2.1 in the Appendix.
>
> **W7: "Some presentations need optimization. For example, Figure 1 is somewhat cluttered and a bit hard to interpret, while the headers in Tables 2 and 3 appear to be misaligned."**
>
> Thanks for your advice. We have improved the structure of Figure 1 and modified Table 2 and 3.

---

### Official Review · Reviewer_sAMU · 2025-11-02

**Soundness:** 3
**Presentation:** 2
**Contribution:** 3
**Rating:** 4
**Confidence:** 5

**Summary:**

This paper introduces an agentic retrieval framework with reasoning augmentation, enabling the base model to learn how to better retrieve evidence from external knowledge bases and perform reasoning for medical visual question answering.
Different from previous text-only agentic retrieval systems that simply copy or rephrase the question to expand the search, the proposed med-rwr first observes visual clues and then generates a multimodal query. to build such a system, the authors curate a dedicated dataset including think, query, retrieve, and answer items, and train the model using a composite reward. Experiments on multiple datasets show consistent performance improvements.

**Strengths:**

1. the proposed method is solid and supported by comprehensive experimental settings and ablation studies.
2. the target problem it aims to address, agentic retrieval and multimodal information fusion, is important for the medical analysis domain.

**Weaknesses:**

Unclear hyperparameter design: the reward function is composite, but the weights assigned to each component vary widely without sufficient explanation or justification. It’s unclear how these weights were determined or whether any sensitivity analysis was performed.

Inappropriate RAG baselines: Although the authors compare their approach with a training-free RAG setup (see Figure 3), they don’t clearly specify what the base model is (is it Med-RWR, and is it based on qwen or lingshu?) In addition, they only compare against two general-domain agentic retrieval methods (Visual-ARFT and MMSearch-R1), but not against medical-domain RAG methods such as MMed-RAG (ICLR ’25) or RULE (EMNLP ’24).
Comparing purely reasoning-based models is also not very meaningful here, since retrieval introduces additional information by design, and the general-domain retrieval/reasoning models are not trained on medical data.

Insufficient generalization validation: While the authors introduce ECBench, it appears to be private, making it hard for others to verify or fully understand the dataset details. To support the generalization claims, I recommend adding experiments on at least one public medical benchmark.

**Questions:**

I think the motivation behind confidence-driven test-time compute scaling is that the retrieved information isn’t always sufficient. So a straightforward fix would be to just retrieve more text chunks, basically scale up retrieval.

But the current setup instead uses another knowledge base for multimodal retrieval. I’m not totally sure I understand the reasoning here. could you help explain the logic? also, why not just use both databases directly at the same time, instead of doing it in two stages?

---

> ### Author Response · Authors · 2025-11-25
> **Response to Reviewer sAMU [Part I]**
>
> We are grateful for your recognition of our paper's performance and potential academic impact. Thanks for your efforts in the review and valuable comments.
>
> ## **Weaknesses**
>
> ### **W1: "Unclear hyperparameter design..."**
>
> We thank the reviewer for raising this insightful question.
>
> **Clarification on Hyperparameter Settings.** Our hyperparameter settings are carefully designed under two principles: (1) **prioritizing the dominant task objective**, which is the accuracy reward guided by $w_a$ and (2) **balancing the scale of other supporting rewards** with $w_f, w_q, w_c$ to ensure stable training. Specially, we intentionally use a smaller weight for Confidence Gain Reward to prevent the model from overly optimizing on the confidence scores, but instead focus on the retrieved information quality. Appendix Figure 8 shows that our chosen weights lead to a stable training process, where all reward components are rescaled into a similar and reasonable range.
>
> **Sensitivity Analysis.** We further conducted sensitivity analysis at the table below. It clearly shows that our final setting achieves the best performance compared to other weight configurations, demonstrating that prioritizing accuracy while maintaining balanced reward scales is crucial for stable and effective training. We have included the sensitivity analysis in Appendix Section E.5.
>
> | **Setting** | **$w_f$** | **$w_a$** | **$w_q$** | **$w_c$** | **MedXpertQA-MM** | **MMMU-H&M** | **MMMU-Pro-H&M** | **ECBench** |
> | --- | --- | --- | --- | --- | --- | --- | --- | --- |
> | All Same Weights | 1 | 1 | 1 | 1 | 23.4 | 57.6 | 37.4 | 43.1 |
> | Low Accuracy Weight | 1 | 1 | 0.4 | 0.5 | 25.1 | 64.8 | 40.9 | 47.1 |
> | High Query Semantic Weight | 1 | 5 | 1 | 0.5 | 25.5 | **66.0** | 41.9 | 49.6 |
> | High Confidence Gain Weight | 1 | 5 | 0.4 | 1 | 26.2 | 62.8 | 41.3 | 50.0 |
> | Ours | 1 | 5 | 0.4 | 0.5 | **27.2** | 65.5 | **43.7** | **51.1** |

---

> > ### Author Response · Authors · 2025-11-25
> > **Response to Reviewer sAMU [Part II]**
> >
> > ### **W2: "Inappropriate RAG baselines..."**
> >
> > We are appreciative for your careful review of the experiment setting and insightful comment regarding the baseline methods.
> >
> > **Clarification of Training-Free RAG Implementation Details.** Firstly, we would like to clarify that the base model used in the "training-free RAG" setup in Figure 3 (b) is the vanilla **Qwen2.5-VL-7B** model. We have stated this detail in Section 4.3.1 of the revised version.
> >
> > **Motivation of Baseline Selection.** The key novelty for performance enhancement of Med-RwR is two-facet: incorporating **retrieval** during **reasoning**. **Retrieval** provides extra knowledge for the model to analyze, while **reasoning** drives the model to thoroughly analyze the input. Therefore, we benchmarked against both reasoning methods and retrieval methods to quantify the advancement brought by these two key characteristics comprehensively.
> >
> > **Improvement in comparison with RAG baselines.** We acknowledge that comparison with medical-domain RAG methods would strengthen our evaluation and have added these experiments.
> >
> > 1. **Comparing against medical-domain multimodal RAG methods on modality-specific VQA benchmarks.** MMed-RAG [1] and RULE [2] are medical multimodal RAG methods that used **image-based retrieval** with **modality-specific vision-language retrievers.** To fairly compare with their methods, we train our model with the same finetuning data and evaluate on the same test benchmarks as MMed-RAG [1] and RULE [2]. The results are shown below (Accuracy as the metric). Results of MMed-RAG and RULE are referenced from their original paper. The results demonstrated that our proposed reasoning-with-retrieval paradigm still has better performance over strong medical RAG baselines under identical experimental setup. We have updated Section E.6 in the revision to incorporate these discussions and experiments.
> >
> >
> >     | **Model** | **IU-Xray** | **MIMIC-CXR** | **Quilt-1M** | **PMC-OA Pathology** |
> >     | --- | --- | --- | --- | --- |
> >     | RULE [2] | 87.84 | **83.92** | 68.97 | 61.41 |
> >     | MMed-RAG [1] | 89.54 | 83.57 | 72.95 | 64.54 |
> >     | Med-RwR | **92.81** | **83.92** | **75.74** | **67.35** |
> > 2. **Comparing with additional medical RAG methods.** We appreciate the reviewer's concern that our performance improvement over reasoning-based models might stem from simply introducing additional information rather than our reasoning-with-retrieval design. Additionally, the reviewer notes that comparison with general-domain agentic retrieval models is limited by their lack of medical pretraining. To address both concerns with a fair ablation, we implemented an additional medical RAG baseline using Lingshu, a generalist medical MLLM **pretrained on various medical data**, augmented with **the same retriever and knowledge source** as our proposed Med-RwR but follows **a standard RAG approach**. The RAG query formulation settings (Retrieve with Question/Refined Query/Image Caption) are the same as the training-free RAG setup explained in Section 4.3.2 and Figure 3 (b). Results are shown in the table below. Med-RwR's performance highlights that training the model to learn when and how to retrieve information is more effective than simply augmenting the medically-pretrained model via  external knowledge. This demonstrates the mutual benefit between retrieval and reasoning: reasoning-guided queries retrieve more relevant information, which in turn leads to more accurate reasoning process and outcome. We have updated the results at Section 4.3.2 in the revised version.
> >
> >
> >     | **Model** | **MMMU-Pro-H&M (4opt)** | **MMMU-Pro-H&M (10opt)** | **MedXpertQA-MM** | **ECBench** |
> >     | --- | --- | --- | --- | --- |
> >     | Lingshu | 50.0 | 37.1 | 26.7 | 44.6 |
> >     | Lingshu+Retrieval with Question | 55.6 | 37.1 | 26.9 | 50.2 |
> >     | Lingshu+Retrieval with Refined Query | 56.6 | 38.1 | 24.9 | 49.9 |
> >     | Lingshu+Retrieval with Image Caption | 54.2 | 37.1 | 25.9 | 49.8 |
> >     | Lingshu+Med-RwR | **57.7** | **42.7** | **28.5** | **51.9** |
> >
> > [1] Xia, Peng, et al. "Mmed-rag: Versatile multimodal rag system for medical vision language models." *The Thirteenth International Conference on Learning Representations*. 2025.
> >
> > [2] Xia, Peng, et al. "Rule: Reliable multimodal rag for factuality in medical vision language models.” *Proceedings of the 2024 Conference on Empirical Methods in Natural Language Processing.* 2024.

---

> > > ### Author Response · Authors · 2025-11-25
> > > **Response to Reviewer sAMU [Part III]**
> > >
> > > ### **W3: "Insufficient generalization validation..."**
> > >
> > > Thanks for your valuable suggestions.
> > >
> > > **Clarification on Evaluated Benchmarks.** We would like to clarify that we have conducted experiments on **multiple public medical VQA benchmarks** like MMMU-H&M, MMMU-Pro-H&M, and MedXpertQA-MM, where our method achieves substantial improvements compared with both reasoning and retrieval baselines, represented by 3.7% improvement on MMMU-Pro-H&M. However, existing public benchmarks cannot evaluate model generalizability on critical medical domains that are **severely underrepresented in training data**. To address this gap, we introduced ECBench, a benchmark on echocardiography, which comprises less than 2% of both public training corpora and our own training data (see Figure 4). Evaluation on this dataset demonstrates that our model can achieve an improved accuracy by dynamically retrieving relevant knowledge from domain-specific knowledge database despite not being specifically trained on echocardiographic data. We will release ECBench under an appropriate copyright license upon acceptance to enable reproducibility.
> > >
> > > **Additional Experiments on Public Dataset.** Following your suggestion, we have conducted additional experiments on [Quilt-VQA](https://huggingface.co/datasets/wisdomik/Quilt_VQA) [1], a public pathology-specific VQA benchmark, to evaluate our model’s generalizability more effectively. Accuracy is applied as the evaluation metric. Notably, our model achieves strong performance on this specialized domain dataset without any domain-specific finetuning, relying solely on proactively retrieving additional information from the external knowledge base. We have included the experiments on this public medical benchmark in Table 1 in the revised version.
> > >
> > > | **Model** | **Quilt-VQA** |
> > > | --- | --- |
> > > | Visual-ARFT | 48.4 |
> > > | MMSearch-R1 | 42.9 |
> > > | MedRegA | 65.9 |
> > > | HuatuoGPT-Vision | 59.5 |
> > > | MedGemma | 65.6 |
> > > | Lingshu | 56.9 |
> > > | MedVLM-R1 | 56.9 |
> > > | Med-R1 | 56.0 |
> > > | Chiron-o1 | 58.0 |
> > > | Qwen-VL (base) | 59.4 |
> > > | Med-RwR (ours) | **67.9** |
> > > | Med-RwR+CDIR (ours) | **69.7** |
> > >
> > > [1] Seyfioglu, Mehmet Saygin, et al. "Quilt-llava: Visual instruction tuning by extracting localized narratives from open-source histopathology videos." Proceedings of the IEEE/CVF Conference on Computer Vision and Pattern Recognition. 2024.

---

> ### Author Response · Authors · 2025-11-25
> **Response to Reviewer sAMU [Part IV]**
>
> ## **Questions**
>
> ### **Q1: "...a straightforward fix would be to just retrieve more text chunks, basically scale up retrieval. But the current setup instead uses another knowledge base for multimodal retrieval...could you help explain the logic?**
>
> We appreciate your insight in the design rationale. We agree with your understanding, while we would like to clarify our motivation to introduce multimodal case retrieval instead of scaling up text retrieval.
>
> **Limitation of scaling up text retrieval.** We have conducted experiments to visualize the performance trend as the number of retrieved text chunks increase. Results are shown below. **Bold** values indicate the best results, and '*' indicates the second-best results. It can be observed that the performance initially rises then declines as the number of retrieved text chunks increases. This can be attributed to the fact that directly retrieving more texts exposes the model to information less similar to the query, which would potentially confuse the model with noisy information.
>
> | **Retrieval Method for Test-Time Scaling** | **MMMU-H&M** | **MMMU-Pro-H&M** | **MedXpertQA-MM** |
> | --- | --- | --- | --- |
> | Retrieve 1 Text Chunk | 63.5 | 40.2 | 26.0 |
> | Retrieve 2 Text Chunks | 64.8 | 41.3 | 27.2* |
> | Retrieve 3 Text Chunks | 65.5* | 43.7* | 27.2* |
> | Retrieve 4 Text Chunks | **66.2** | 40.9 | 26.8 |
> | Retrieve 5 Text Chunks | 61.4 | 39.5 | 25.4 |
> | **Retrieve Multimodal Case (Our CDIR Method)** | **66.2** | **44.1** | **27.5** |
>
> **Advantage of multimodal case retrieval over scaling up text retrieval.** Our two retrieval databases serve as distinct and complementary roles: **general knowledge database** provides broad and fundamental medical knowledge, and **multimodal** **medical case database** supplements targeted patient-specific cases. While the general knowledge database is comprehensive for most medical knowledge, there are challenging corner cases that cannot be covered by this database, requiring complementary information from practical examples. Our confidence-driven method can detect these potential corner cases and address them by retrieving similar cases from the medical case database. An example is shown in Figure 13 and Figure 14, where a retrieved multimodal case provides relevant information while the general knowledge base fails. This is further supported by the last row of the table above, which shows that additionally incorporating CDIR achieves better performance than retrieving from the textual KB alone. The performance gain can be attributed to the benefit of image re-retrieval, which enables the model to reference visual evidence from similar patient cases when the textual knowledge base alone is insufficient. Thus, CDIR represents an effective test-time scaling method by leveraging multimodal cases rather than expanding text retrieval.
>
> ### **Q2: "Also, why not just use both databases directly at the same time, instead of doing it in two stages?"**
>
> Thanks for your careful review and critical questions. We introduce Confidence-Driven Image Re-retrieval as a second-stage conditional enhancement after the proactive textual retrieval due to following considerations:
>
> 1. **Computational efficiency.** Since general knowledge database have already contains broad medical knowledge, the initial retrieval from this source can readily support analysis in most of the cases. Retrieving from general knowledge database only introduces low computation by comparing text chunks, while retrieving from medical case database needs to comparing similarity between image features, requiring more computational costs than textual retrieval given higher data dimensions and denser information. Therefore, we selectively apply image re-retrieval to candidates with lower confidence, which reduces the overall computational load while still providing more diverse knowledge for uncertain samples.
> 2. **Mimicking clinical workflow.** This sequential retrieval design inherently mimics the diagnostic process in the real-world scenario. Typically, doctor first refers to general guidelines, which mirrors our proactive textual retrieval from the medical knowledge base. When doctors find guidelines inadequate for confident decision-making, they also consult multimodal records from similar patient cases. Our proposed Confidence-Driven Image Re-retrieval is analogous to this process.

---

### Author Response · Authors · 2025-11-25
**General Response**

We sincerely appreciate the reviewers for their time and effort in the review. We are glad that they highlighted that our research problem `important for the medical analysis domain`/`critical and high-stakes`/`sound` (sAMU, Wkgq, Hige) and found our proposed method `solid` /`demonstrate the superior performance`/`novel` (sAMU, fZCw, Wkgq). We have carefully addressed all reviewer comments and revised the manuscript accordingly. A revised version has been uploaded with all changes marked in blue.

---

### Author Response · Authors · 2025-12-02
**Summary**

Dear AC and SAC,

We sincerely thank your continuous and additional effort in evaluating our paper. We would like to take this opportunity to provide a brief summarization and clarification for the rebuttal process to facilitate your assessment, as the reviewing situation has changed and the deadline is approaching.

All four reviewers seem to appreciate our paper, agreeing that our **motivation** to enhance trustworthiness of medical reasoning MLLMs through proactively retrieving external factual knowledge `is important for the medical analysis domain` (**sAMU**) and `critical and high-stakes` (**Wkgq**). The key contribution, **Reasoning-with-Retrieval framework,** was acknowledged as `well-motivated` (**Wkgq**), `sound` (**Hige**), and `solid` (**sAMU**). They mentioned that the framework’s **reward design** is `a key strength` , appraising it as `a nice way to optimize for the utility of the retrieved information` (**Wkgq**). All reviewers also have consensus for the `comprehensive` (**sAMU, Wkgq**) experiment with `superior performance` (**fZCw, Hige**). Additionally, reviewers agreed that the paper was `well-written` and `clear` (**fZCw, Hige**).

Notably, reviewers are open for discussion at the initial stage, mentioning that they will `consider raising my score`(**Wkgq**) / `be willing to adjust my ratings` (**Hige**) if we can `address the limitations in weaknesses` (**Wkgq**) / `address the concerns` **(Hige)**. The reviewers’ remaining concerns mainly centered around the design motivations, implementation details, and additional evaluations. We have carefully addressed these concerns with point-by-point responses and have revised the manuscript accordingly. We provide a brief summarization for our rebuttal as follows.

### **Common Concerns**

- Comparison with Multimodal RAG Methods in Medical Domain (**sAMU, Hige**)
    - We add new comparisons with Medical RAG methods, demonstrating that our proposed reasoning-with-retrieval paradigm still has better performance over strong medical RAG baselines under identical experimental setup. This highlights the advantage of our proactive retrieval paradigm over standard medical RAG methods.
- Sensitivity Analysis of Hyperparameter (**sAMU, Hige**)
    - We clarify the underlying principles for our hyperparameter selection and present sensitivity analysis, which proves the validity of our hyperparameter design.
- Motivation and Scalability of the proposed Confidence-Driven Image Re-retrieval (CDIR) mechanism for test-time scaling (**sAMU, Wkgq, Hige**)
    - We explain the motivation behind the CDIR mechanism design, which is aimed to augment low-confidence inferences with patient-specific cases.
    - We empirically prove the effectiveness of CDIR over alternative scaling approaches which simply increase the number of retrieved textual chunks.
    - We demonstrate the scalability of the CDIR mechanism, showing that its performance is further strengthened as the retrieved corpus expands, and discuss its potential to scale to larger multimodal corpora.

### **Specific Concerns**

- Additional Generalization Evaluation (**sAMU**)
    - We clarify the motivation of evaluation benchmark choices and validate generalizability on an additional modality-specific benchmark, where our method achieves the best performance through proactively retrieving the external KB without domain-specific finetuning.
- Clarification on Methodological Design, Implementation Details, and Experimental Setup (**fZCw**)
    - We provide detailed explanations on reward function design choices, curriculum learning implementation, and base model specifications.
- Presentation Improvement (**fZCw**)
    - We improve figure and table presentation to enhance readability.
- Multi-hop Reasoning Capacity (**Wkgq**)
    - We demonstrate our multi-turn capability with a concrete example in the current framework.
    - We show that single-turn retrieval already achieves substantial improvements for our targeted general medical scenarios.
    - We discuss future extensions for robust multi-hop reasoning in complex cases such as rare diseases.
- Discussion of Recent Related Methods (**Wkgq**)
    - We add discussion about the most recent retrieval-based methods and reasoning-with-retrieval methods, highlighting our contribution as the first to integrate the reasoning-with-retrieval pipeline within medical MLLMs and bridge a critical gap in previous studies.

Thank you again for your considerations and efforts in assessing our paper.

Best Regards,

Med-RwR Authors

---

### Meta-Review · Area_Chair_Wpei · 2026-01-01

**Summary:**

This paper proposes Med-RWR, a multimodal reasoning-with-retrieval framework for medical VQA that trains models to proactively retrieve external knowledge during reasoning using reinforcement learning. While all reviewers acknowledged the importance of the problem and the solid experimental setup, significant concerns remain regarding the novelty over existing medical RAG methods, the gap between claimed multi-turn retrieval capabilities and actual single-turn behavior, and the scalability of the proposed CDIR mechanism. The reviewers converged on scores at the borderline (4, 6, 4, 6), reflecting uncertainty about whether the contributions are sufficient for acceptance.

**Reviewer Concerns:**

Addressed concerns:

Hyperparameter sensitivity (sAMU-W1): The authors provided a comprehensive sensitivity analysis demonstrating that their chosen weights achieve optimal performance. This concern is adequately resolved.

Baseline comparisons (sAMU-W2, Hige-W3): The authors added comparisons with MMed-RAG and RULE under identical experimental setups, showing improvements. Additional ablations with Lingshu+RAG further demonstrate that gains are not solely from retrieval. This concern is largely addressed.

Public benchmark evaluation (sAMU-W3): Experiments on Quilt-VQA were added, showing strong generalization. This concern is resolved.

Related work discussion (Wkgq-W4): The authors incorporated discussions of recent RAG and reasoning-with-retrieval methods. This concern is resolved.

Outstanding concerns:

Multi-turn retrieval capability (Wkgq-W3): This remains the most critical unresolved issue. The paper claims support for multi-turn reasoning, but the authors acknowledge that only 0.4-1.3% of cases actually use multiple turns. The failure case in Figure 15 demonstrates that when initial retrieval fails, the model cannot recover. The authors' response that "single-turn is sufficient for most scenarios" does not adequately address the discrepancy between claimed and actual capabilities.

CDIR scalability (Wkgq-W2): The authors argue that scaling to larger corpora is "straightforward" using FAISS, but no experiments beyond 50k images were conducted. The claim that this is merely an engineering effort rather than a methodological concern is not fully convincing, as scalability is essential for practical deployment.

Conceptual novelty: Multiple reviewers implicitly questioned whether the contribution is sufficiently novel given the extensive prior work on medical RAG. The core idea of "proactive retrieval during reasoning" is incremental, and the performance improvements over existing medical RAG baselines (MMed-RAG, RULE) are modest.

**Reviewer Scores:**

Reviewer sAMU (initially 4): Would likely remain at 4. The authors addressed the hyperparameter and baseline concerns well, but the core contribution remains incremental.

Reviewer fZCw (initially 6): Would likely remain at 6. Most concerns were clarifications that were adequately addressed.

Reviewer Wkgq (initially 4): Would likely remain at 4 or reduce to 2. The critical concerns about multi-turn retrieval and CDIR scalability were not substantively resolved. The authors acknowledged these as limitations rather than providing solutions.

Reviewer Hige (initially 6): Would likely remain at 6.

---

### Decision · Program_Chairs · 2026-01-26

Reject